



# Future snow changes and their impact on the upstream runoff in Salween

Chenhao Chai[1,2], Lei Wang[1,2*], Deliang Chen[3], Jing Zhou[1], Hu Liu[1,2], Jingtian Zhang[1,2], Yuanwei Wang[4], Tao Chen[5]

[1] State Key Laboratory of Tibetan Plateau Earth System, Resources and Environment (TPESRE), Institute of Tibetan Plateau Research, Chinese Academy of Sciences, Beijing 100101, China

[2] The University of Chinese Academy of Sciences, Beijing, China

[3] Department of Earth Sciences, University of Gothenburg, Gothenburg, 40530, Sweden

[4] School of Geographical Sciences, Nanjing University of Information Science & Technology, Nanjing, China

[5] School of Geography and Planning, Sun Yat-Sen University, Guangzhou 510275, China

* **Corresponding to**: Lei Wang, e-mail: wanglei@itpcas.ac.cn

**Abstract:** Understanding the hydrological processes related to snow in global mountainous regions under climate change is necessary for achieving regional water and food security (e.g., the United Nation's sustainable development goals (SDGs) 2 and 6). However, the impacts of future snow changes on the hydrological processes in the high mountains of the Third Pole are still largely unclear. In this study, we aimed to project future snow changes and their impacts on the hydrology in the upstream region of Salween (US) under the SSP126 and SSP585 emission scenarios using a physically-based cryosphere–hydrology model. We found that in the future, the climate would become warmer (SSP126 ($0.2°C \cdot 10$ yr$^{-1}$); SSP585 ($0.7°C \cdot 10$ yr$^{-1}$)) and wetter (SPP126 (5 mm 10 yr$^{-1}$) and SSP585 (27.8 mm 10 yr$^{-1}$)) in the US under these two shared socioeconomic pathways (SSPs). Under this context, the snowfall, snow cover, snow water equivalent, and snowmelt runoff are projected to exhibit significant decreasing trends during 1995–2100, and the decreases are projected to be most prominent in summer and autumn. The future (2021–2100) snowmelt runoff is projected to significantly increase in spring compared to the reference period (1995–2014), which would benefit the availability of water resources in the growing season. The annual total runoff would significantly increase in all the future periods due to increased rainfall, which would increase the availability of water resources within the basin, but the high peak flow that occurs in summer may cause rain flooding with short



duration and high intensity. Compared to the reference period (the contribution of snowmelt runoff to the total runoff was determined to be 17.5%), the rain–snow-dominated pattern of runoff would shift to a rain-dominated pattern after the near term

(2021–2040) under SSP585, while it would remain largely unchanged under SSP126. Climate change would mainly change the pattern of the snowmelt runoff, but it would not change the annual hydrograph pattern (dominated by increased rainfall). These findings improve our understanding of the responses of cryosphere–hydrological processes under climate change, providing valuable information for integrated water

resource management, natural disaster prevention, and ecological environmental protection at the Third Pole.

**Keywords**: Climate change, Future snow change, Cryosphere-hydrology model, Runoff, Salween.

## 1. Introduction

Snow, a key component of the cryosphere, is widely distributed in high mountain regions around the world, which are particularly sensitive to climate change; thus, it is an important indicator of regional climate change (Nepal et al., 2021; Pulliainen et al., 2020). The snowpack can store a large amount of solid precipitation in the cold season and can melt in the warm season. Accordingly, it not only has a strong effect on the

regional hydrological cycle (Huning and AghaKouchak, 2020; Musselman et al., 2021) but also provides abundant water resources for all walks of life in basins and supports about one-sixth of the world's population (Yan et al., 2022; Barnett et al., 2005). In addition, the snow cover (SC) affects the radiation balance and the thermal regime of the underlying ground due to the high albedo of snow, and thereby, it changes the

regional energy balance, which in turn affects the regional climate (Jia et al., 2021; You et al., 2020; Henderson et al., 2018; Xiao et al., 2017). Changes in snowfall, snow storage, and snowmelt under climate change would not only change the total annual and seasonal runoff at mountain outlets but would also affect and change mountain glaciers and the availability of water resources in downstream regions (Immerzeel et



al., 2020; Li et al., 2019). These factors may increase the frequency, intensity, and range of natural disasters and may further threaten the security of the water supply, flood/drought control, and ecological security in downstream areas (Qin et al., 2020; Biemans et al., 2019). Therefore, an in-depth understanding of the impact of climate change on the snow-related hydrological regime in mountainous areas is crucial to regional ecological protection, water resource management, disaster prevention, and sustainable socio-economic development (Nepal et al., 2014; Biemans et al., 2019; Yao et al., 2019; Tang et al., 2019; Viviroli et al., 2020; Qi et al., 2020).

Over the last few decades, the Tibetan Plateau (TP) has experienced intense warming, which has exceeded the global warming rate (IPCC, 2019). In addition, the precipitation has also exhibited an increasing trend in the central TP and a decreasing trend in the southern and eastern TP under climate warming (Chen et al., 2015). Moreover, several studies have predicted that the TP would continue to warm in the future, accompanied by increased precipitation, especially in the monsoon-controlled regions, with increased frequency and intensity of extreme events (Panday et al., 2015; Sanjay et al., 2017). These influences not only change the spatiotemporal distribution and magnitude of the precipitation but also alter several of the key variables (e.g., solid precipitation) that drive snow occurrence and development, as well as the variables (e.g., radiation and temperature) that control snow ablation (Hock et al., 2019). Yao et al. (2019) and Bibi et al. (2018) reported that the melting of the snowpack, especially in low- and mid-elevation areas (because more of the precipitation occurs as rainfall), has accelerated in recent decades on the TP, which has increased the instability and uncertainty of the inter-annual and seasonal runoff and would cause future changes in the spatiotemporal pattern and availability of water resources in the TP river basins (Tang et al., 2019; Immerzeel et al., 2010; Kraaijenbrink et al., 2021). Although many studies have assessed several snow variables (e.g., snowfall, snow storage, SC, and snowmelt) and the hydrological processes related to snow under climate change on the TP based on in-situ observations and land surface snow/hydrological models (Bian et al., 2020; Xu et al., 2017; Barnhart et al., 2016; Su et al., 2016), such as conceptual



models (e.g., day–degree approach), physically-based distributed snowmelt runoff

models, and hydrological models coupled with snow processes (e.g., Variable Intiltration Capacity model (VIC) and Spatial Processes in Hydrology model (SPHY)), the impact of future snow changes on runoff is still unclear due to a lack of reliable data. Moreover, most of these models did not fully consider the physical processes of snow accumulation and ablation (Wang et al., 2017; Liu et al., 2018). Therefore, a

comprehensive cryosphere–hydrological model and high-quality forcing datasets are urgently needed to improve our understanding of snow-related hydrological processes on the TP to better support the sustainable development of this region.

To better understand the effect of snow changes on the TP on runoff under climate change, in this study, the upstream region of Salween (US) was selected as the research

object. The US is located in the alpine region of the TP and Hengduan Mountains, which has a complex underlying surface and is very sensitive to climate change (Liu et al., 2017; Chen et al., 2020). Moreover, as a transboundary river with great influence, the reasonable utilization and coordination of water resources in the Salween Basin have received a great deal of attention from relevant countries and organizations worldwide

(Yao et al., 2012). Previous studies have shown that the underlying surface, ecosystem, and hydrological processes of the basin have undergone intense changes in recent years, which have gradually changed the total water resources and runoff changes within the basin, further affecting the rational allocation of water resources and hydropower development (Liu et al., 2017; Fan and He, 2012; Luo et al., 2017; Hong and He, 2019).

However, the working conditions in the US are very difficult due to the complexity of the topography and the harsh environment, which has resulted in the distribution of national meteorological stations in the basin being irregular and relatively sparse. In addition, due to a lack of continuous snow-hydrological observations in the basin resulting from strict national data policies, it is difficult to quantify and reveal the

mechanisms of the snow-related hydrological processes and verify the results of numerical simulations (Wang et al., 2021). Moreover, the precipitation and snow products used in previous studies have low accuracy and poor applicability in this basin



and are often insufficient for analysis of trends due to the short periods of the records. These factors have seriously restricted our understanding of the snow-related hydrological processes and the development and utilization of water resources in this basin (Mao et al., 2019; Liu et al., 2016; Ding et al., 2015).

Previous studies have used simple statistical methods to study the variations in the historical runoff in the US based on observation data from individual sites; the dry season and annual runoff have been found to exhibit an increasing trend as a result of the increase in precipitation and meltwater (glacier and snow), wherein the precipitation contributes the most (Zhang et al., 2007; You et al., 2008; Yao et al., 2012; Cuo et al., 2014; Luo et al., 2016; Liu et al., 2017; Zhang et al., 2019). However, these studies did not separate glacier and snowmelt, consider the corresponding physical processes, or predict future changes in the runoff and its components. In addition, Su et al. (2016), Lutz et al. (2014), Zhao et al. (2019), and Khanal et al. (2021) used hydrological models to predict the future runoff on the TP, and they all found that the future runoff would exhibit an increasing trend, but there are great uncertainties in the meltwater contribution and seasonal variations in these models. This may be due to the different descriptions of the cryospheric processes used in these models and the large differences in the driving datasets used. Moreover, these studies did not consider the intermediate snow change processes, which may lead to a partial understanding of the snow-hydrological processes.

The main objective of this study was to simulate the changes in the snow-related hydrological processes in the US on the southeastern TP, China. First, based on meteorological observation data, we evaluated the performances of four reanalysis precipitation products and selected the most reliable products. Second, we constructed a distributed cryosphere–hydrological model of the US basin (WEB-DHM-sf; Wang et al., 2009a, b, 2010, 2016, 2017) and evaluated its performance using observation discharge and remote sensing data (land surface temperature (LST), moderate resolution imaging spectroradiometer (MODIS) SC). Third, we used the WEB-DHM-sf driven by global climate model (GCM) CMIP6 to predict the changes in the snowfall,



snow cover, snow water equivalent (SWE), total snowmelt, snowmelt runoff, and total runoff during different periods under different shared socioeconomic pathway (SSP) scenarios, and we further analyzed the impact of the snow cover changes on the runoff.

## 2. Study Area

The Salween River, also known as the Nujiang River in China, is a free-flowing international river, and it is also an important strategic hydropower and water resource reserve area in Southeast Asia (Lu et al., 2021). It originates from the southern foot of the Tanggula Mountains on the Tibetan Plateau in China. It flows across Myanmar and Thailand from north to south and finally flows into the Andaman Sea (He, 2004). In China, its length and basin areas are 2013 km and 137,800 km², respectively (Guo, 1985). The basin above Jiayuqiao (JYQ) is called the US, with an average elevation and area of 4800 m and 72975 km², respectively (Fig. 1a). This region is a typical alpine mountain zone and is controlled by the westerlies and South Asian monsoon. The precipitation is mainly concentrated from May to October (wet season). The temperature and precipitation are highly dependent on elevation and exhibit significant spatial heterogeneity (Mao et al., 2019). Therefore, this region is also sensitive to climate change (Liu et al., 2017). The snow depth exhibited a significant decreasing trend in the US during 2000–2018. The maximum monthly snow depth occurred in December–January, and the lowest monthly snow depth occurred in July–August (Yan et al., 2021). The annual mean fraction of the snow cover was about 50% during 2001–2014, and it was mainly distributed at 4000–6000 m (Li et al., 2017). The area covered by glaciers was greater than 800 km², accounting for 1.2% of the entire basin. The soil was mainly lithosol soil, followed by frozen soil, and a small amount of planosol soil (Fig. 1b). The vegetation types were mainly agriculture/C3 grassland, with some shrubs (Fig. 1c).

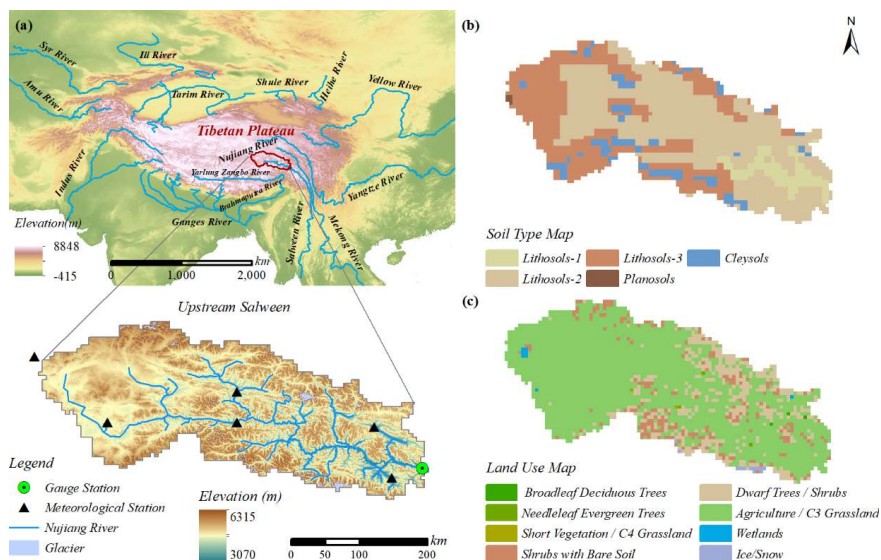

Figure. 1 (a) Topography and distribution of meteorological and hydrological stations, (b) soil type, and (c) land use types in the upstream region of the Salween (US) Basin.

## 3. Data and Methods

### 3.1 Distributed cryosphere–hydrological model

The WEB-DHM-sf has been further improved based on the first version of the distributed biosphere hydrological model (WEB-DHM; Wang et al., 2009a, b), which coupled and improved the three-layer energy balance snow module of the simplified simple biosphere 3 model (SSib3) (Shrestha et al., 2010) and the empirical frozen soil parameterization scheme (Wang et al., 2010). The three-layer snow module divides the snow layer in the grids with snow depths of greater than 5 cm into three layers; otherwise, it is regarded as single-layer snow. It considers the energy exchange between the snow layers and the influences of the incidence angle of the solar radiation and the age of the snow on the snow albedo. Therefore, this model can describe the physical processes of snow in detail, including the phase transition, compaction, albedo, temperature, and melt runoff of each layer. Thus, we can obtain the SC, snowfall, SWE, snowmelt, and other variables from the model outputs (Shrestha et al., 2014). The frozen ground processes are characterized by the frozen soil hydrothermal transfer



parametrization scheme, in which the thermal conductivity scheme used is the Johansen

scheme (Wang et al., 2017). In both snow parameterization schemes, enthalpy is used

instead of temperature to establish the energy equation or as a new predictive variable,

which reduces the uncertainty when calculating the latent heat released by the changes

in the water phase and enhances the stability of the model (Wang et al., 2017; Song et

al., 2020). In this version of the model, glaciers are considered to be snow with a

thickness of 100 m (the details of this model have been described by Wang et al. (2016)

and Shrestha et al. (2010, 2014)). With the continuous improvements in recent years,

this model can better describe the cryospheric-hydrological processes in the alpine

region, and it has been verified to have a good performance in several basins in the TP

(Liu et al., 2018; Qi et al., 2019; Qi et al., 2022; Zhong et al., 2020; Wang et al., 2021;

Zhou et al., 2022), especially in areas where observations are very scarce. In addition,

the outputs of the model can be verified using multi-source data, such as in-situ

observations and satellite remote sensing data. Therefore, this model was used in this

study to analyze the impact of the snow changes on the runoff with a temporal resolution

of 1 hour and a spatial resolution of 5 km.

### 3.2 Data

The meteorological forcing data included the near-surface air temperature, total

precipitation, downward shortwave and longwave radiation, wind speed, surface

pressure, and specific humidity. For the historical period (1995–2014), except for the

precipitation data, the data for the meteorological variables were obtained from the

China Meteorological Forcing Dataset (CMFD), which has a high spatiotemporal

resolution (3 hours and 0.1°) from 1979 to 2018 and has been widely used in

hydrometeorological research because of its good applicability (He et al., 2020). The

precipitation data were obtained from the ECMWF 5th generation reanalysis

data product (ERA5). We selected four widely used precipitation products, including

the CMFD, Global Land Data Assimilation System (GLDAS), ERA5, and Multi-

Source Weighted-Ensemble Precipitation (MSWEP), which is from Beck et al. (2017),



and we verified these products using observations. Finally, based on this evaluation, we selected the ERA5 for use in this study. The evaluation results are shown in Figs. S1

and S2. We selected the ERA5 based on its 1 hour and 0.25° resolutions from 1979–present, its better spatiotemporal distribution, and the fact that it has been widely used in hydrological simulations of high mountain regions (Hersbach et al., 2020; Yang et al., 2021). The projected climate forcing variables used in the hydrological model were obtained from the Inter-Sectoral Impact Model Intercomparison Project 3b (ISIMIP3b)

(Lange and Büchner, 2021), of which four global climate models (GCM: GFDL-ESM4, IPSL-CM6A-LR, MPI-ESM1-2-HR, MRI-ESM2-0) that have a consistent experimental protocol (historical, SSP126, and SSP585) and atmospheric climate variables (spatial and temporal resolutions of 1 day and 0.5°) were selected as the model input. The ISIMIP3BASD v2.4.1 method was used for the bias adjustment and

statistical downscaling of these variables to ensure their consistency with the long-term statistics of the observation reference dataset, i.e., W5E5 v2.0 (WFDE5 v0.1 over land merged with ERA5 over the ocean) (Lange 2019, 2021a, 2021b; Cucchi et al., 2020). To maintain the relative and absolute trends of these variables during the historical and future periods at the basin scale, we further corrected these variables on the monthly

scale using the delta method, which has been widely used and can effectively avoid a large model bias (Cuo et al., 2011; Su et al., 2016). The historical and future meteorological forcing data were interpolated to the model's resolution (5 km) via bilinear interpolation. The time downscaling method used for the daily forcing data was consistent with that used by Song et al. (2020).

The digital elevation model (DEM) was used to extract the basin's boundary and calculate the topographic parameters. The DEM was the National Aeronautics and Space Administration (NASA) Shuttle Radar Topography Mission (SRTM) DEM with a spatial resolution of 90 m. The soil type and land use maps with a spatial resolution of 1 km used to calculate the soil hydraulic parameters and land use type were obtained

from the Food and Agriculture Organization (FAO, 2003) and the U.S. Geological Survey (USGS) (Figs. 1a, b), respectively. The fraction of the photosynthetically active



radiation (FPAR) and the leaf area index (LAI) used to calculate the dynamics of the vegetation were downloaded from the Global Land Surface Satellite (GLASS) datasets with temporal and spatial resolutions of 8 days and 0.05° (Xiao et al., 2014). The second

glacier inventory of China was provided by the National Tibetan Plateau Data Center (Guo et al., 2015). These data were resampled to the model's resolution (5 km).

The daily observed discharge data at the outlet (JYQ) of the US basin were obtained from the National Hydrology Almanac of China and were used to calibrate and validate the hydraulic parameters of the hydrological model during 1981–1987.

There are six meteorological stations within the basin. Among them, Naqu and Dingqing stations are the international exchange stations (Figure 1a). The meteorological observation data used to evaluate the precipitation products were obtained from the China Meteorological Administration (CMA). The land surface temperature (LST) data were obtained from the MODIS and MOD11A2 products, with

spatial and temporal resolutions of 1 km and 8 days from 2001 to 2018 (Wan et al., 2014). The SC data were obtained from an improved Terra-Aqua MODIS snow cover and Randolph Glacier Inventory 6.0 combined product (MOYDGL06*) for high-mountain Asia with temporal and spatial resolutions of 8 days and 500 m from 2002 to 2018, which can be accessed from PANGAEA (Sher and Amrit, 2019).

**3.3 Model Evaluation Criteria**

The evaluation criteria used to evaluate the performance of the hydrological model outputs (e.g., discharge, LST, and snow cover) mainly included the Nash-Sutcliffe coefficient (NSE), Kling-Gupta coefficient (KGE), correlation coefficient (CC), mean bias (MB), root mean square error (RMSE), and relative bias (RB). The NSE and KGE

can complement each other and make the evaluation more reasonable (Gupta et al., 2009). The equations used to calculate these evaluation criteria are as follows:

$$NSE = 1 - \frac{\sum_{i=1}^{n}(Sim_i - Obs_i)^2}{\sum_{i=1}^{n}(Obs_i - \overline{Obs})^2}, \tag{1}$$



$$KGE = 1 - \sqrt{(CC-1)^2 + \left(1 - \frac{\delta_{sim}}{\delta_{obs}}\right)^2 + \left(1 - \frac{\overline{Sim}}{\overline{Obs}}\right)^2}, \qquad (2)$$

$$CC = \frac{\sum_{i=1}^{n}(Sim_i - \overline{Sim})(Obs_i - \overline{Obs})}{\sqrt{\sum_{i=1}^{n}(Sim_i - \overline{Sim})^2}\sqrt{\sum_{i=1}^{n}(Obs_i - \overline{Obs})^2}}, \qquad (3)$$

$$MB = \frac{\sum_{i=1}^{n}(Sim_i - Obs_i)}{n}, \qquad (4)$$

$$RMSE = \sqrt{\frac{\sum_{i=1}^{n}(Sim_i - Obs_i)^2}{n}}, \qquad (5)$$

$$RB = \frac{\sum_{i=1}^{n}(Sim_i - Obs_i)}{\sum_{i=1}^{n}Obs_i} \times 100\%, \qquad (6)$$

where $Sim_i$ is the simulated value at time $i$; $Obs_i$ is the observed values at time $i$; n is the number of samples; $\overline{Sim}$ and $\delta_{sim}$ are the mean value and standard deviation of the simulated values, respectively; and $\overline{Obs}$ and $\delta_{obs}$ are the mean value and standard deviation of the observed values, respectively. The closer the values of NSE, KGE, and CC are to 1, the better the simulation. The closer the values of MB, RMSE, and RB are to 0, the better the simulation.

## 4. Results

### 4.1 Model calibration and validation

#### 4.1.1 Model calibration and validation using observed discharge

First, the model parameters were calibrated using the daily observed discharge data measured at JYQ station from 1981 to 1983, which mainly included the following soil hydraulic parameters: the saturated soil moisture content ($\theta_s = 0.48$), residual soil moisture content ($\theta_r = 0.07$), hydraulic conductivity anisotropy ratio ($anik = 2.17$), Van Genuchten parameter ($\alpha = 0.014, n = 2.385$), saturated hydraulic conductivity for soil surface ($K_{sat1} = 141.45 \, mm \cdot h^{-1}$), saturated hydraulic conductivity for unsaturated areas ($K_{sat2} = 21.22 \, mm \cdot h^{-1}$), and hydraulic conductivity for an



unconfined aquifer ($K_g = 0.7\ mm \cdot h^{-1}$). Then, the daily observed discharge data for

1984–1987 (observations for 1986 were missing) were used to validate the calibrated

hydrological model. The results are shown in Fig. 2. The time series of the simulated

discharge matched the observations during 1981–1987 well and can capture the daily

and monthly changes in the observed discharge, especially in the dry season. The values

of the evaluation criteria during the calibration and validation period were determined

to be as follows: 0.7 (NSE), 0.75 ($R^2$), 0.84 (KGE), 3.18% (RB), and 0.79 (NSE), 0.81

($R^2$), 0.88 (KGE), 3.29% (RB) (Fig. 2a). The monthly simulated and observed discharge

were in better agreement during the peak flow (Fig. 2b), accurately reproducing the

inter-annual and intra-annual variations in the discharge at JYQ station, although there

were overestimates and underestimates during the wet seasons in some years.

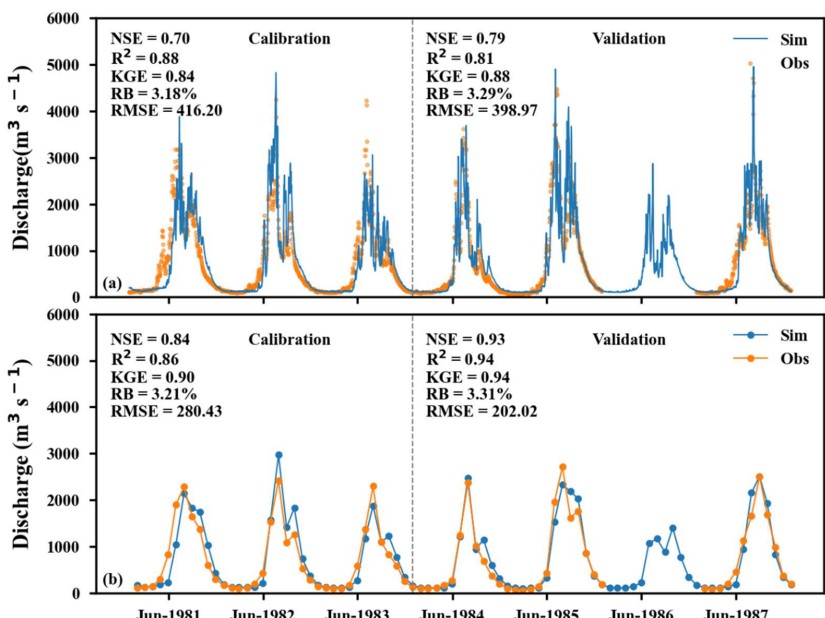

Figure. 2 Simulated and observed (a) daily and (b) monthly discharges at Jiayuqiao (JYQ) station
from 1981 to 1987. The calibration and validation periods were 1981−1983 and 1984−1987,
respectively.

**4.1.2 Model validation using LST**

Due to the lack of observed discharge data from 1988 to the present, remote





sensing data (MODIS LST and snow cover) were used to further validate the
performance of the hydrological model at the basin scale. Figure 3 shows the
comparison of the results for the basin-averaged time series between the simulated and
MODIS LST in the daytime (10:30, local time) and nighttime (22:30, local time) during
2001–2018. The results show that the simulated LST is in good agreement with the

MODIS LST in the daytime ($R^2 = 0.69$) and nighttime ($R^2 = 0.91$). The simulated
nighttime LST has a higher $R^2$ value and lower MB value, but the RMSE was better
than that of the daytime LST, which indicates that the LST was mainly affected by the
incoming solar radiation and land surface absorption during the daytime and was
affected very little at night.

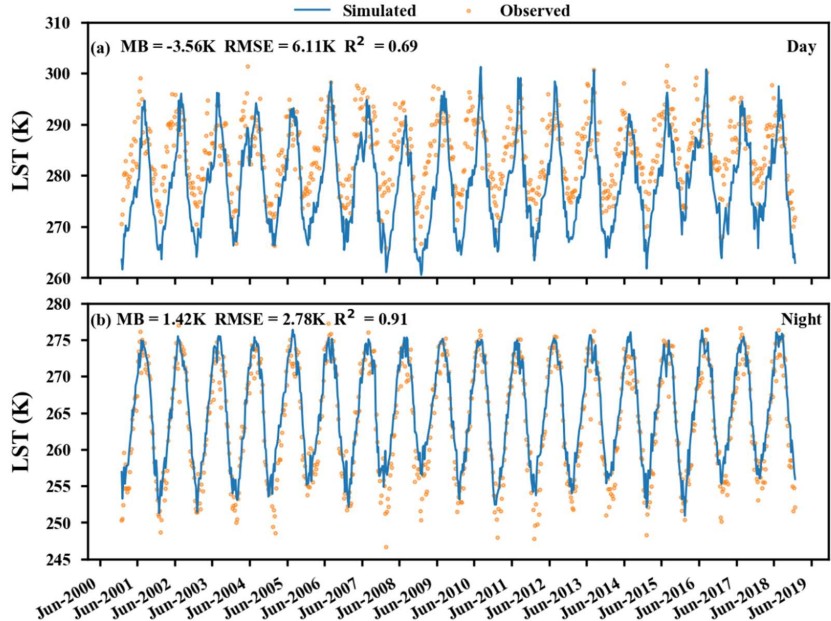


Figure. 3 Simulated (a) daytime and (b) nighttime 8-day land surface temperature (LST) compared
to MODIS satellite observations, averaged over the US basin during 2001−2018.

From the perspective of the spatial distribution patterns (Fig. 4), the simulated
multi-year seasonal average LST during the daytime and nighttime were similar to those

of the MODIS LST from 2001 to 2018. Similarly, the simulated nighttime LST was
better than the simulated daytime LST. The simulated seasonal LST during the

nighttime was very close to the MODIS value in terms of the spatial distribution and

magnitude, while the simulated seasonal LST during the daytime was significantly

underestimated in winter, spring, and autumn and was slightly overestimated in spring.

The reason for this difference may be that the LST results were closely related to the

temperature, and the very limited CMA stations in the US were used to generate the

CMFD data. This may also have introduced some uncertainty in the temperature lapse

rate, which was used in the interpolation of the CMA observations to produce the

CMFD air temperature. The uncertainties of these forcing data would lead to

uncertainties in the simulated LST.

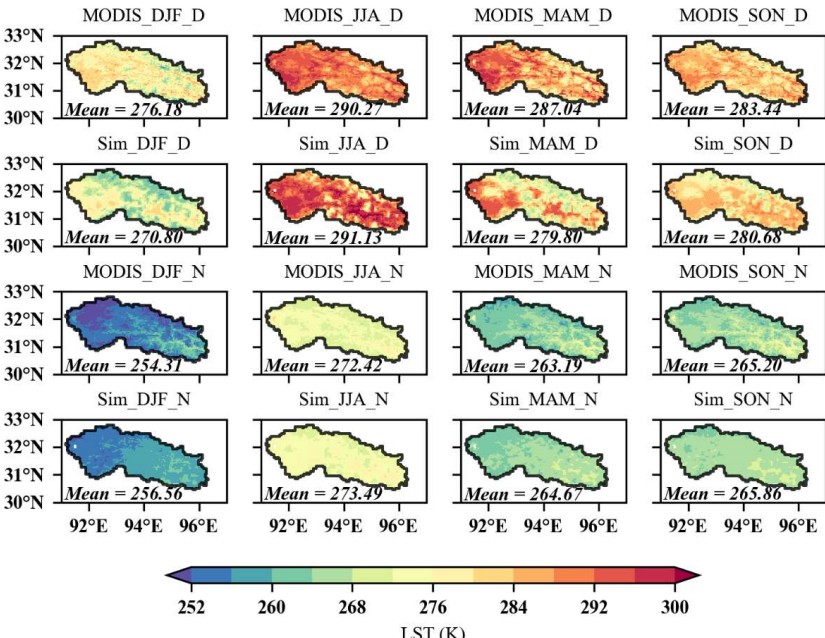

Figure. 4 Comparison of the spatial distributions of the land surface temperature (LST) simulation
and MODIS data for the US basin from 2001 to 2018, in the daytime (first and second rows) and
nighttime (third and fourth rows). From left to right the panels are winter (DJF), summer (JJA),
spring (MAM), and autumn (SON).

### 4.1.3 Model validation using MOYDGL06* SC

The fraction of simulated snow cover area (FSCA) was also verified using the

MOYDGL06* FSCA data at the basin scale from 2003 to 2019 at the 8-day and

monthly scales (Fig. 5). As is shown in Fig. 5a, there was a good correlation between



the simulated 8-day FSCA and the MOYDGL06* 8-day FSAC (CC = 0.73). It can be

seen from Fig. 5b that the simulated FSCA was overestimated, except for in October

and November, especially in summer (by almost 100%). Of course, this could reflect

the intra-annual changes in the MOYDGL06* FSCA values at the basin scale. Figure

5c illustrates the good agreement between the simulated and MOYDGL06* FSCA on

the monthly scale during 2003–2018 ($R^2$ = 0.56), which generally reflects the inter-

annual changes in the MOYDGL06* FSCA in the US basin.

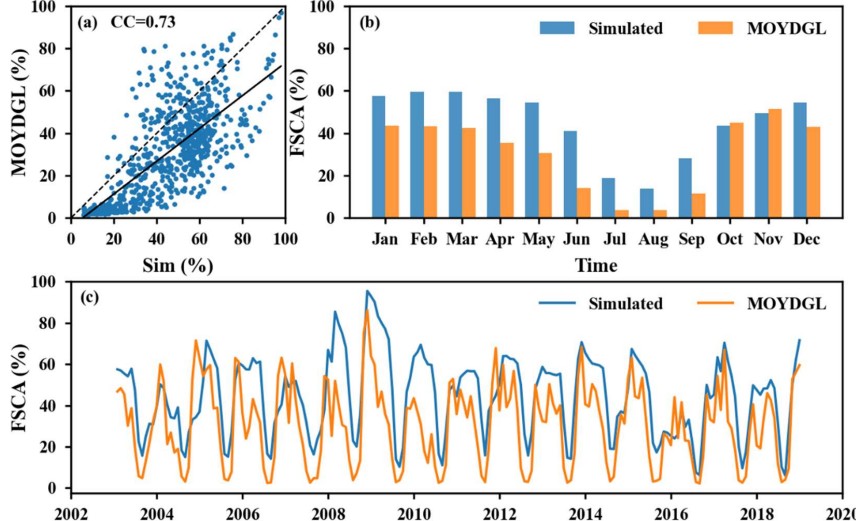

Figure. 5 Comparison of the simulated fraction of the snow cover area (FSCA) and the
MOYDGL06* time series in the US basin. (a) 8-day FSCA; (b) multi-year mean monthly FSCA;
and (c) variations in the monthly FSCA during 2003–2018.

As can be seen from Figs. 6 and S3, the simulated SC reproduced the spatial

distribution of the seasonal evolution of the SC in the US basin, and it also captured the

snow ablation (April–August) and accumulation (September–March) processes well.

Moreover, the simulated SC also reflected the rapid melting of snow after June 2 and

the rapid accumulation of snow after October 8, but there was a difference on some

days, that is, the SC was overestimated in the high mountains in summer and

underestimated in the low valleys in winter. If the uncertain grids in the MOYDGL06*

were considered (red grids), the difference would be smaller. The differences in some

months may partly be caused by the bias in the forcing data, especially the biases in the





temperature and precipitation. In addition, the resolution of the MOYDGL06* (500 m)
is not consistent with that of the hydrological model (5 km), which may also lead to
differences in the calculation of the average SC for the entire basin.

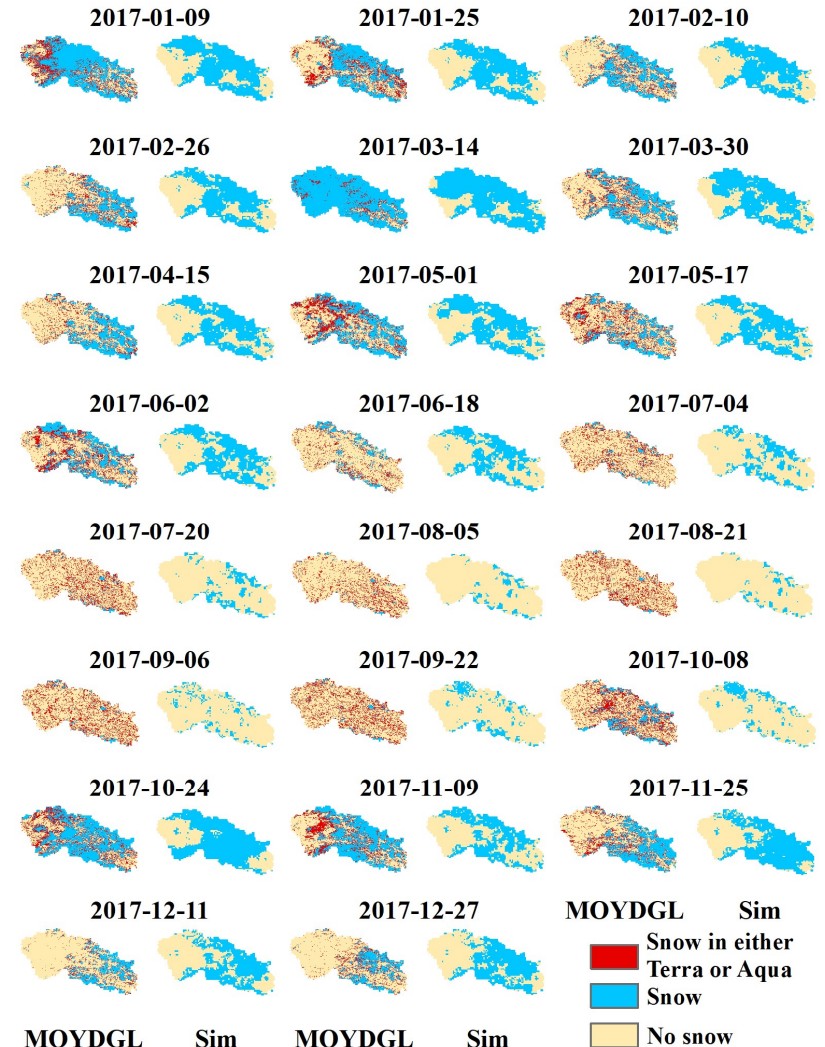

Figure. 6 Comparison of the MODIS 8-day snow cover product (MOYDGL06*) and the simulated
8-day snow cover in the US basin in 2017. The time interval is 16 days.

Overall, the verification results show that the simulation results of the hydrological
model are reasonable and can be used to simulate the historical and future snow-
hydrological processes in the US basin.



## 4.2 Future climate change

### 4.2.1 Evaluation of projected data

Due to the systematic bias and coarse spatial resolution of the GCM, there was great uncertainty in using it directly to drive the hydrological model at the basin scale (Zhang et al., 2016; Liu et al., 2018). Thus, it was necessary to evaluate these data before conducting the hydrometeorological analysis. Four GCM datasets with bias adjustment conducted using the delta method were used as the meteorological forcing data to drive the WEB-DHM-sf hydrological model. The results are shown in Fig. 7. Although the simulated values of the single GCM in the wet season were overestimated (GFDL-ESM4, IPSL-CM6A-LR) or underestimated (MPI-ESM1-2-HR, MRI-ESM2-0), the daily simulated discharge of each GCM was relatively consistent with the inter-annual variations of the observations during 1981–1987 (Fig. 7a). Moreover, the daily hydrograph curve of the multi-GCM ensemble mean (MEM) reproduced the discharge at JYQ well, with reasonable NSE (0.61), KGE (0.77), $R^2$ (0.64), and RB (-9.62%) values. The monthly hydrograph curve of the MEM and observed data exhibit a good overall agreement (NSE = 0.78, KGE = 0.77, $R^2$ = 0.79, and RB = -9.7%) during 1981–1987 (Fig. 7b), and the intra-annual changes in the discharge at JYQ station were captured. In summary, the MEM was better than the single GCM, and thus, the MEM was used in the analysis of the snow-related hydrological processes in the US basin.

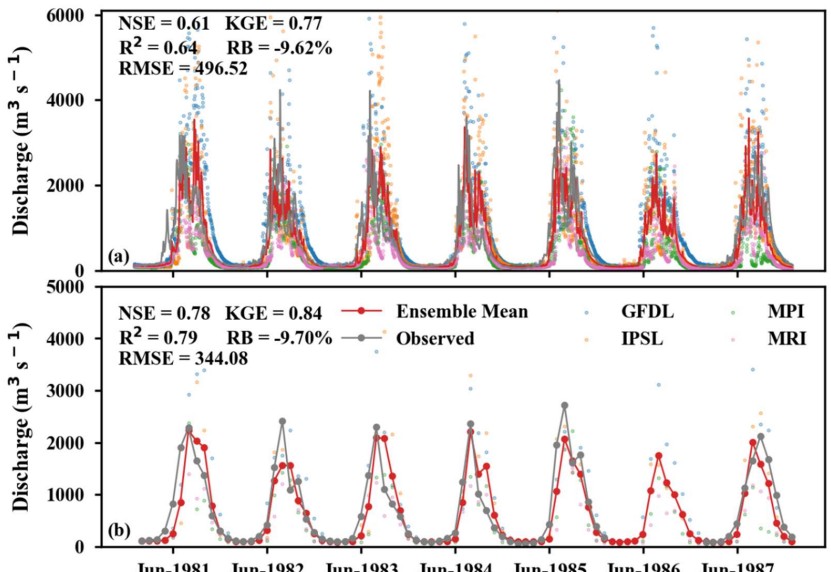

Figure. 7 Comparison of simulated discharge of four GCMs and the observed discharge at JYQ
station during 1981–1987. The red line is the multi-GCM ensemble mean discharge, and the gray
line is the observed data on the (a) daily and (b) monthly time scales.

#### 4.2.2 Projected changes in temperature and precipitation

As is shown in Table S1, the temperature would exhibit significant growth trends

under SSP126 (0.2°C • 10 yr$^{-1}$) and SSP585 (0.7°C • 10 yr$^{-1}$) during 1995–2100. It also

shows a consistently increasing trend during the different periods, except for in the long

term for SSP126 for which a slightly significant decrease occurs. In addition, the

precipitation would also exhibit growth trends throughout the entire study period under

SPP126 (5 mm 10 yr$^{-1}$) and SSP585 (27.8 mm 10 yr$^{-1}$), and it would also exhibit an

increasing trend in the near term, mid-term, and long term, except for during the

reference period when it exhibited an obvious decreasing trend (SSP126: −5.8 mm 10

yr$^{-1}$, SSP585: −9 mm 10 yr$^{-1}$). The climate in the US would generally undergo a

warming and wetting trend in the future. For the different periods, both the warming

and wetting trend of the basin would gradually slow down under SSP126, while they

became much stronger under SSP585.

Figure 8 and Table S2 show the relative changes in the annual precipitation and

mean annual temperature in the US during 1995–2100, with 1995–2014 as the baseline, under SSP126 and SSP585. Compared with the reference period, the warming amplitude would increase in the near, mid- and long-term periods, but the rate of increase would slow down and would not exceed 2°C at the end of this century under

the SSP126. However, the warming exceeds 2°C in the future in the mid-term and would reach about 6°C by the end of this century under SSP585 (Fig. 8a). The annual precipitation would increase by 4.64–5.57% under SSP126 and by 1.34–26.99% under SSP585 during the different periods (Fig. 8b).

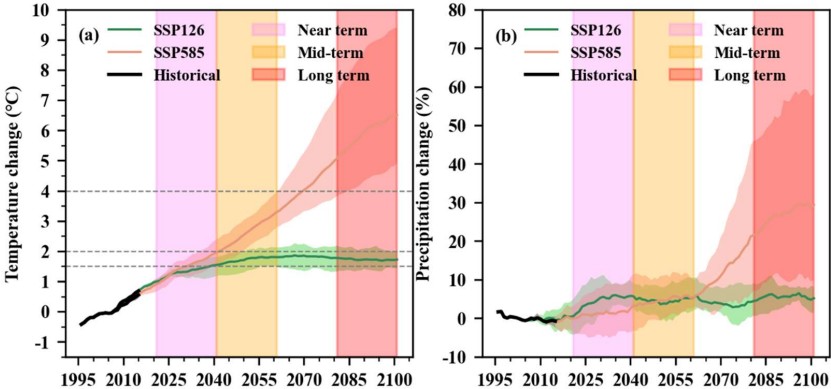

Figure. 8 Relative changes in the annual precipitation and mean annual temperature from 1995 to 2100. The black, red, and green lines represent the precipitation and temperature during the reference period (1995–2014) and under CMIP6 SSP585 (2015–2100) and SSP126 (2015–2100), respectively. The rectangular shaded areas are the near-term (pink: 2021–2040), mid-term (yellow: 2041–2060), and long-term (red: 2081–2100) periods. The shading around the lines represents the

fluctuation range of the data, and the upper and lower ranges are 95% and 5%.

The seasonal cycles of the projected temperature and precipitation in the US would exhibit different changes (Fig. 9). Intense warming would occur in all of the seasons under SSP126 and SSP585. Compared with the reference period, the changes in the seasonal warming rates for each period would be relatively small under SSP126, but

these changes would be greater under SSP585, with a distinct gradient. The warming rates during each season would not exceed 2°C in the near term, and the smallest change would occur in spring under all of the scenarios; however, it would exceed 4°C in the long term under SSP585 (Figs. 9a, b). As can be seen from Figs. 9c and 9d, the

precipitation in the other periods compared to the reference period is projected to

increase from April to September and decrease from October to March under all of the

SSPs, and its growth amplitude would gradually increase from the near term to the long

term. The above results indicate that the US is a typical monsoon region, which would

become warmer–wetter in the monsoon season and warmer–dryer in the non-monsoon

season in the future.

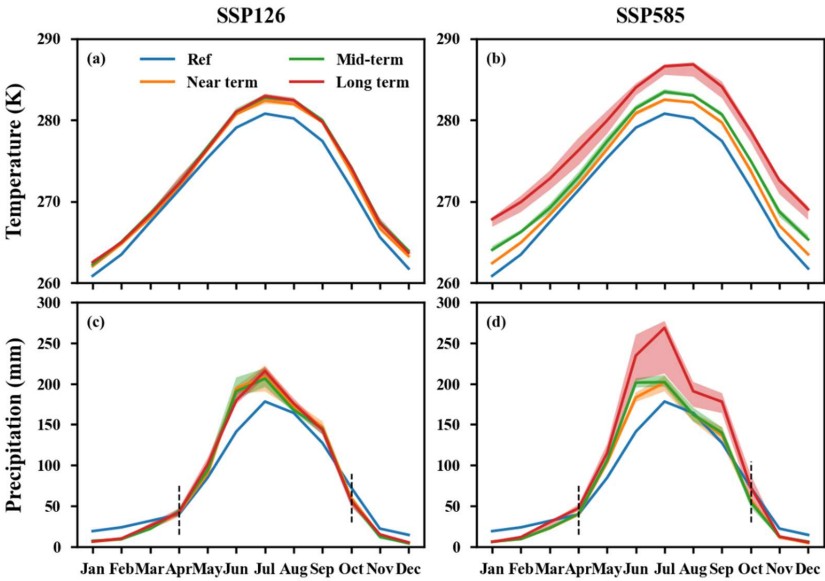


Figure. 9 Seasonal changes in the temperature and precipitation in the reference period, the near
term, the mid-term, and the long term under (a, c) SSP126 and (b, d) SSP585. The shading around
the lines represents the fluctuation range of the data, and the upper and lower ranges are 95% and
5%.

**4.3 Snow related hydrological response to climate change**

**4.3.1 Changes in snowfall**

There would be a significant decreasing trend in the annual snowfall and snow–

precipitation ratio (Snow/Pre) in the US under all of the SSPs during 1995–2100 (Figs.

10a, b). The rate of decrease of the snowfall under SSP585 (13.6 mm 10 yr$^{-1}$) would be

much faster than that under SSP126 (4.2 mm 10 yr$^{-1}$). In the near term, the snowfall is

projected to decrease by more than 13% under SSP126 and 16% under SSP585. By the



end of the century, the snowfall would be reduced to about 118 mm under SSP585, which is less than half of the snowfall in the reference period, while the snowfall would decrease by approximately 16% under SSP126. Similarly, the snow–precipitation ratio would exhibit changes similar to those of the snowfall, and the snow–precipitation ratio is projected to decrease to approximately 10% under SSP585 and 20% under SSP126 by the end of the century.

Figures 10c and d show the changes in the monthly scale snowfall and rainfall at the basin scale. The rainfall would mainly occur from May to Oct. (monsoon season), accounting for more than 70% of the total annual precipitation, and the changes in and pattern of the rainfall in the future would be similar to those of the total precipitation. However, the snowfall would exhibit a distinct bimodal pattern, with the first peak appearing in May (accounting for about 21% of the annual snowfall) and the second in Oct. (accounting for about 14% of the annual snowfall). The snowfall is projected to decrease by less than 10% from Nov. to Apr. compared to the reference period, and there would likely be no snowfall in July and Aug. after the mid-term. Moreover, the projected snow–precipitation ratio exhibits a consistent decrease in all of the months compared to the reference period (Figs. 10e, f). Further analysis of the seasonal variations in the snowfall revealed that the snowfall would be the heaviest in spring, and it is projected to decrease by about 30% (1%) under SSP585 (SSP126) by the end of the century. The largest reduction in the snowfall would occur in the summer and autumn under all of the SSPs, and the snowfall is likely to decrease by approximately 85% (44%) and 60% (21%), respectively, under SSP585 (SSP126) by the end of the century. The above results indicate that the precipitation in the US is less likely to occur in the form of snow in the future under climate warming.





Table 1. Changes in and trends of the seasonal snowfall in the US basin during different periods under SSP126 and SSP585 compared to the reference period (1995–2014).

| Item | Period | Spring | | Summer | | Autumn | | Winter | |
|---|---|---|---|---|---|---|---|---|---|
| | | SSP126 | SSP585 | SSP126 | SSP585 | SSP126 | SSP585 | SSP126 | SSP858 |
| **Snowfall (mm)** | Reference | 91.1 | 91.1 | 52.3 | 52.3 | 74.3 | 74.3 | 18.7 | 18.7 |
| | Near term | 89.4 | 89.4 | 34.4 | 32.2 | 63.9 | 59.2 | 16.9 | 16.9 |
| | Mid-term | 86.1 | 82.2 | 26.3 | 24.8 | 57.7 | 49.4 | 17.9 | 17.9 |
| | Long term | 90.2 | 64.4 | 29.3 | 7.6 | 58.5 | 30.2 | 18.1 | 16.1 |
| **Relative change (%)** | Near term | -1.9 | -1.9 | -34.3 | -38.5 | -13.9 | -20.3 | -10.0 | -10.0 |
| | Mid-term | -5.5 | -9.8 | -49.7 | -52.6 | -22.4 | -33.6 | -4.7 | -4.5 |
| | Long term | -0.9 | -29.3 | -44.0 | -85.5 | -21.3 | -59.3 | -3.3 | -14.3 |
| **Trend (mm yr⁻¹)** | Reference | 0.0 | 0.0 | -0.1 | -0.1 | -0.3 | -0.3 | 0.0 | 0.0 |
| | Near term | 0.7 | 0.1 | -0.5 | 0.0 | 0.1 | -0.4 | 0.2 | 0.1 |
| | Mid-term | -0.1 | 0.3 | -0.1 | -0.6 | 0.3 | -0.6 | 0.4 | -0.1 |
| | Long term | 0.7 | -1.9 | -0.2 | -0.5 | 0.5 | -0.4 | -0.1 | -0.1 |

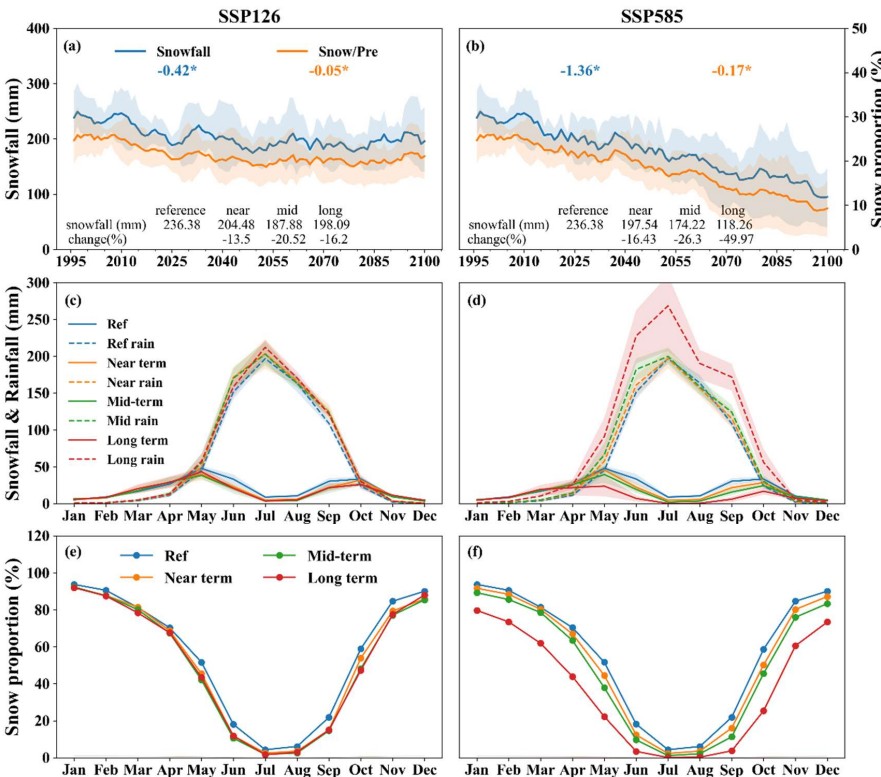

Figure. 10 Snowfall changes and the ratio of snowfall to total precipitation (TP). (a, b) Annual
changes in and proportion of snowfall, (c, d) seasonal changes in snowfall and rainfall, and (e, f) the




ratio of snowfall to total precipitation on the seasonal scales in the US basin during different periods under SSP126 and SSP585 compared to the reference period. The shading around the lines represents the fluctuation range of the data, and the upper and lower ranges are 95% and 5%.

### 4.3.2 Changes in SCA

The trends of and relative changes in the annual FSCA at the basin scale under the different SSPs during 1995–2100 are shown in Fig. S4 and Table S3. The annual FSCA would exhibit a significantly decreasing trend at the basin scale during 1995–2100, and this trend would be more obvious under SSP585 ($-0.42\%$ yr$^{-1}$). Compared with the reference period, the annual FSCA is projected to increase by 4.9% in the near term and

to decrease by 8.59% and 2.77% in the mid-term and long term, respectively, under SSP126. In contrast, the annual FSCA is projected to drastically decrease under SSP585, and it would be less than 40% compared to the reference period by the end of the century. Moreover, it should be noted that the SC would continue to decrease from the near term to the mid-term under SSP126, but it would begin to increase in the long term, which

is significantly different from the pattern under SSP585. This indicates that the future SC in the US would decrease further under climate warming, and the severe warming under SSP585 would cause the SC to melt rapidly.

    The results obtained from the analysis of the monthly scale decadal FSCA from 1995 to the 2090s are shown in Figs. 11a and b. Figure 11 shows that the cycle of

accumulation and ablation of the FSCA would increase gradually from September–March and decreased rapidly from April–August. Under SSP585, the FSCA was projected to decrease in almost all months. However, the FSAC would increase in June–November and decrease in December–April under SSP126. This pattern would also occur for the monthly scale FSCA during the different periods (Figs. 11c, d). Similarly,

the changes in the FSCA would be ere more pronounced under SSP585 than under SSP126. For example, compared to the reference period, the FSCA would change to snow-free from July–September by the end of the century under SSP585.

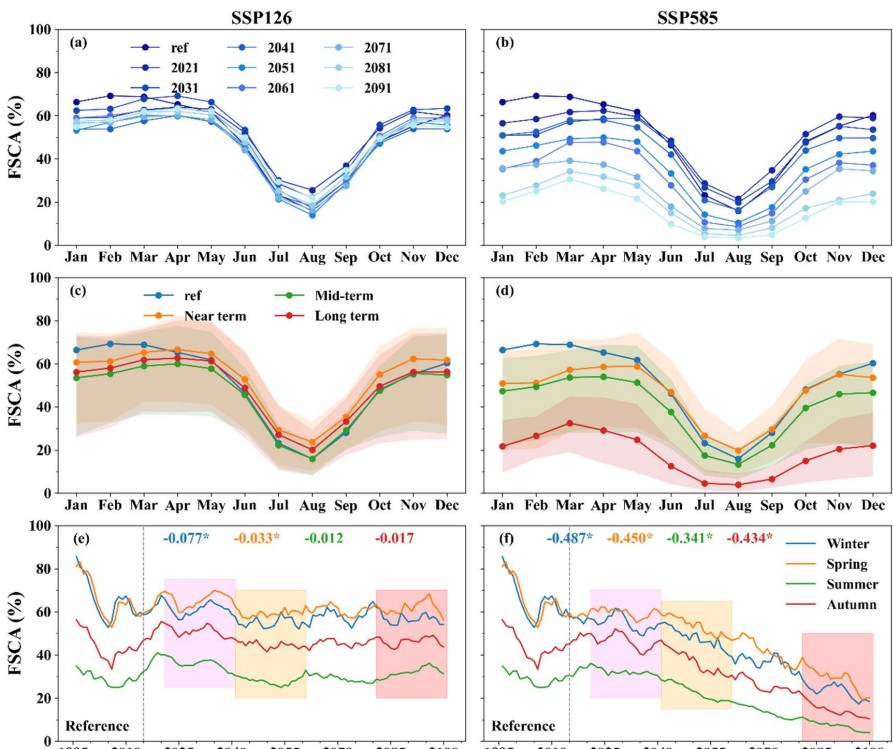

Figure. 11 Changes in the monthly and seasonal FSCA in the US basin during different periods
under SSP126 and SSP585 compared to the reference period: (a, b) Decadal FSCA changes; and (c,
d) FSCA changes in different periods. The shading around the lines represents the fluctuation range
of the data, and the upper and lower ranges are 95% and 5%. (e, f) Variations in seasonal FSCA
during 1995–2100. The number represents the trends of the corresponding line (the same color),
and * indicates significance at $P < 0.05$. The rectangular shaded areas indicate the near term (pink),
mid-term (yellow), and long term (red).

Figures 11e and f and Table S4 present the change trends of the seasonal FSCA,
i.e., in spring (MAM), summer (JJA), autumn (SON), and winter (DJF). The results
show that there would be a decrease during all of the seasons during 1995–2100, but
the decrease would be more significant in winter and spring. Compared to the reference
period, the FSCA is projected to decrease most by about 61% on average in winter and
spring under SSP585. Even under SSP126, the FSCA would decrease by about 15% on
average in winter and spring. However, there are differences between SSP126 and
SSP585 in summer and autumn. From the near term to the long term, the FSCA is
projected to increase in summer and autumn under SSP126, but this would not offset



the large losses in spring and winter. This shows that the reduction of the SC in winter
and spring would cause the reduction of the annual SC, which may further lead to a
decrease in the snow storage in the basin, thus affecting the amount of snowmelt during
the ablation period.

From the perspective of different elevations, the SC would mainly be distributed
at mid-high elevations (≥ 4500 m), accounting for about 44% of the basin area during
the reference period, while less than 7% of the SC would be located at low elevations
(< 4500 m) (Table S3 and Fig. 12). During 1995–2100, the annual FSCA at all
elevations (3500–6000 m) would significantly decrease under all of the SSPs, except in
the area above 5500 m a.s.l. under SSP126, in which no change would occur (Fig. S4).
Further analysis revealed that the rate of decrease of the FSCA would be faster at mid-
high elevations (4500–5500 m) than at low elevations. However, the magnitude of the
decrease in the FSCA would be much greater at low elevations than at mid-high
elevations in all of the periods compared to the reference period. For example, the FSCA
in the low elevation areas is projected to decrease by about 78% under SSP585 and 14%
under SSP126 by the end of the century. This illustrates that the amount of SC lost at
mid-high elevations would be much greater than that lost at low elevations throughout
the 21st century, but the response of the SC at low elevations to climate warming would
be more rapid and pronounced.



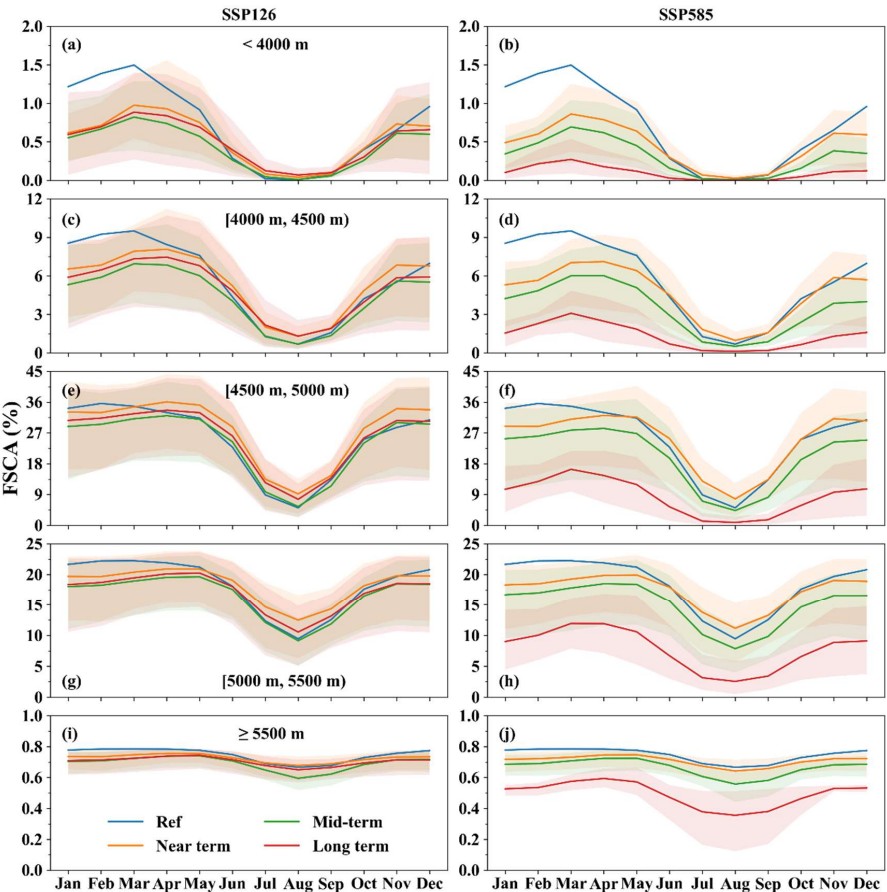

Figure. 12 Change in the monthly fraction of snow cover area (FSCA) in the different elevation bands in the different periods under SSP126 and SSP585 compared to the reference period. The shading around the lines represents the fluctuation range of the data, and the upper and lower ranges are 95% and 5%.

Further statistical analysis of the monthly scale FSCA at all elevations during all of the periods revealed that there would be a consistent decrease in the FSCA at all elevations compared to the reference period (1995–2014), except in the areas with elevations of 4500–5500 m a.s.l. in the near term under SSP126 (Table S3 and Fig. 12). Under all of the SSPs, the FSCA at mid-high elevations (≥ 4500 m) is projected to decrease in almost all months from the mid-term to the long term. At low elevations (< 4500 m), the FSCA would decrease by approximately 25% from Dec. to Apr. under SSP126 by the end of the century, while there would be a slight increase in the FSCA



during June-November (with no snow cover in July and Sept.). However, the FSCA is projected to decrease by approximately 80% in almost all months under SSP585 by the end of the century. Moreover, the decrease in the FSCA would be more pronounced in winter and spring at low elevations than at high elevations under all of the SSPs, and it would further strengthen towards the end of the century. Remarkably, we found that the snowmelt would start earlier at high elevations than at low elevations under SSP585 by the end of the century. This was confirmed at elevations of 4500–5500 m a.s.l., where the starting time of the decrease in the FSCA would shift from May to March, and it would continue until the end of Aug. Correspondingly, the FSCA in this elevation range would also decrease by more than 60%. In addition, there would be almost no snow at low-mid (~5000 m) elevations from July-September, and the snow would be very spare and broken at high elevations. For all of the SSPs, the projected climate would not be conducive to the occurrence and development of SC.

### 4.3.3 Changes in SWE

Figure 13 and Table 2 show the monthly and seasonal scale changes in the SWE in the different periods under SSP126 and SSP585. Compared to the reference period, the SWE is projected to decrease by about 39% in the near term and mid-term, and it would slightly rebound in the long term (about 31%) under SSP126. However, there would be a clear decrease in the SWE from the near term to the long term under SSP585, and the SWE would decrease by more than 85% by the end of the century. Similarly, the characteristics of the above-described changes would also be reflected in the seasonal changes in the SWE (Figs. 13c, d). The SWE in the US would be the highest in spring and the lowest in autumn. Further analysis of the data revealed that the SWE would decrease in all of the seasons and all of the periods under all of the SSPs, and the largest decrease would occur in summer and autumn, with a decrease of more than 91% (30%) under SSP585 (SSP126) by the end of the century. Furthermore, the peak in the SWE would shift from June to May under SSP585 by the middle of the century, and the SWE would be almost zero from July to Sept. However, this would not occur under SSP126. Moreover, it can be seen that the peak in the SWE after the near term would





shift from June to May under SSP585; while it would remain in June under SSP126,

and the peak value of the SWE would weaken after the reference period.

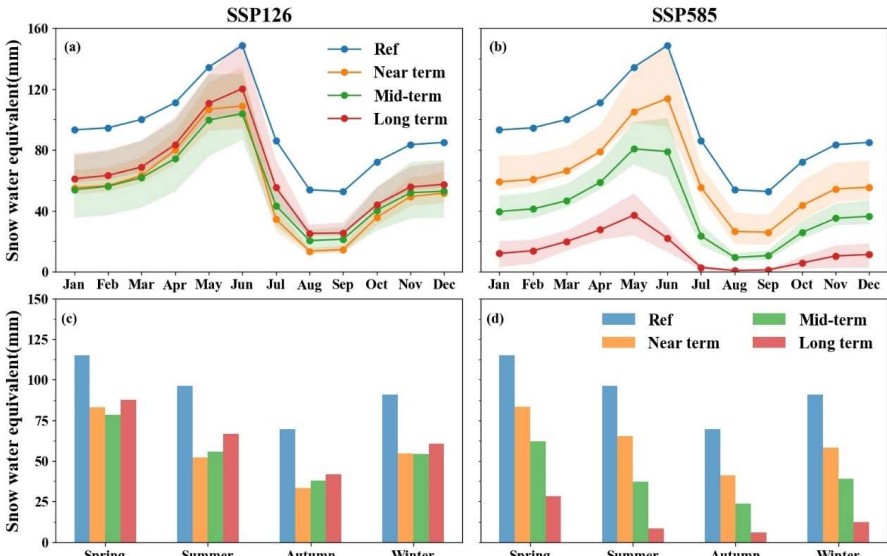

Figure. 13 Seasonal changes in the SWE in the US basin during different periods under SSP126 and
SSP585 compared to the reference period.

Table 2. Changes in the seasonal SWE in the US basin under SSP126 and SSP585 compared to the
reference period (1995–2014).

| Item | Period | Spring | | Summer | | Autumn | | Winter | | annual | |
|---|---|---|---|---|---|---|---|---|---|---|---|
| | | SSP126 | SSP585 | SSP126 | SSP585 | SSP126 | SSP585 | SSP126 | SSP585 | SSP126 | SSP585 |
| **SWE (mm)** | Ref. | 115.15 | 115.15 | 96.27 | 96.27 | 69.48 | 69.48 | 90.89 | 90.89 | 92.95 | 92.95 |
| | Near term | 83.25 | 83.46 | 52.31 | 65.22 | 33.25 | 41.32 | 54.50 | 58.39 | 55.83 | 62.10 |
| | Mid-term | 78.61 | 62.02 | 55.90 | 37.30 | 37.92 | 23.88 | 54.34 | 39.11 | 56.69 | 40.58 |
| | Long term | 87.61 | 28.20 | 66.91 | 8.54 | 41.77 | 5.84 | 60.60 | 12.40 | 64.38 | 13.74 |
| **Relative change (%)** | Near term | -27.70 | -27.52 | -45.66 | -32.25 | -52.14 | -40.53 | -40.03 | -35.76 | -39.93 | -33.19 |
| | Mid-term | -31.73 | -46.13 | -41.93 | -61.25 | -45.42 | -65.63 | -40.21 | -56.97 | -39.00 | -56.34 |
| | Long term | -23.91 | -75.51 | -30.50 | -91.13 | -39.88 | -91.60 | -33.33 | -86.36 | -30.74 | -85.21 |

### 4.3.4 Changes in snowmelt and the snowmelt runoff

The trends of the annual total runoff, total snowmelt, and snowmelt runoff during

1995–2100 are shown in Fig. S5. The total runoff would increase under all of the SSPs,

and it would be faster under SSP585 (4.4 mm yr$^{-1}$) than under SSP126 (0.31 mm yr$^{-1}$).





However, the total snowmelt, including the snowmelt runoff and snowmelt that seeps into the soil, would significantly decrease under SSP585 (−1.04 mm yr⁻¹) and SSP126

(−0.34 mm yr⁻¹). The snowmelt runoff would change very slightly and insignificantly under SSP126, while it would significantly decrease (0.56 mm yr⁻¹) under SSP585. During the reference period, the snowmelt runoff (0.1 mm yr⁻¹) and its contribution to the total runoff (0.3 mm yr⁻¹) significantly increased, but the total runoff exhibited a nonsignificant downward trend (−0.13 mm yr⁻¹). The trends of the total runoff,

snowmelt runoff, and its contribution in the mid-term and long term would be opposite to those during the reference period under all of the SSPs, except in the long term under SSP126 (Figs. 14i–l and 15i–l). This indicates that the snowmelt and snowmelt runoff would likely peak in the near term (2021–2041).

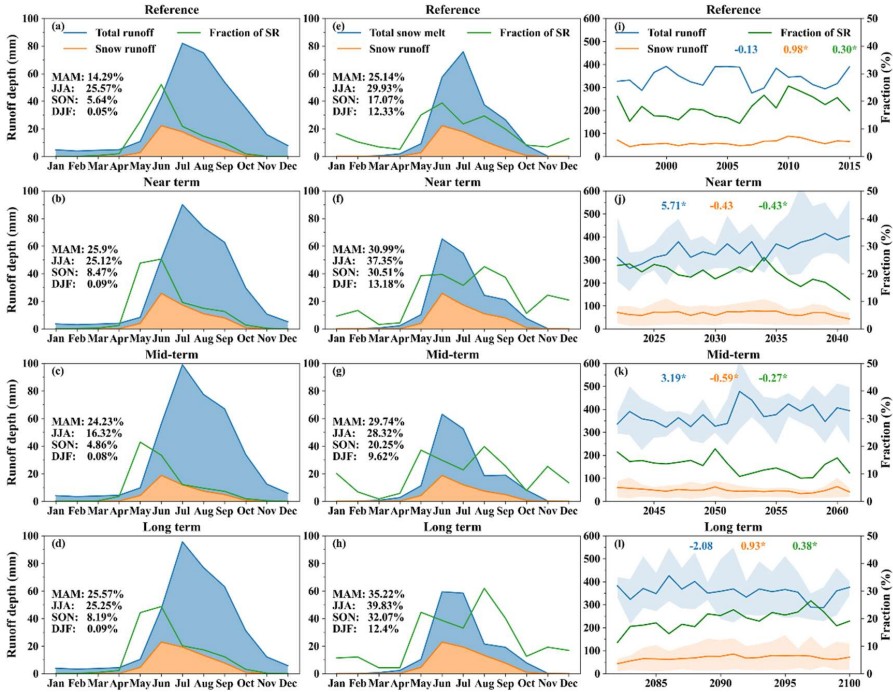

Figure. 14 Changes in the snowmelt in the US basin during the different periods under SSP126 compared to the reference period; (a–d) Comparison of snowmelt runoff (SR) and total runoff at the basin outlet; (e–h) Comparison of snowmelt runoff and total melt (snowmelt runoff plus snowmelt that seeps into the soil); (i–l) Changes in the total runoff and snow meltwater during the different periods. The number represents the trend of the corresponding line (the same color). The text in the

figure is the proportion of the snowmelt runoff in each season to the total runoff (total snowmelt) in the corresponding season, and * indicates a significant trend.





Table S5 presents the changes in the annual total runoff, total melt, and snowmelt runoff from the near term to the long term under SSP126 and SSP585. In the near term, the total runoff would slightly increase under SSP126 (1.07%) and SSP585 (1.40%). In

the mid-term, the total runoff would increase by twice as much under SSP585 (22.86%) compared to under SSP126 (10.4%). By the end of the century, the total runoff is projected to increase by 112.29% under SSP585 compared to the reference period (341.0 mm), whereas there would only be a small increase under SSP126 (4.7%). The total snowmelt would consistently decrease from the near term to the long term under

all of the SSPs. The reduction in the snowmelt would remain below 20% under SSP126, while it would decrease by 22.5% and 43.01% in the mid-term and the long term under SSP585, respectively. The snowmelt runoff would increase in the near term and long term and decrease in the mid-term (by −20.3%) under SSP126, while the snowmelt runoff would decrease significantly from the near term to the long term under SSP585,

and the projected decrease is about 75% in the long term.

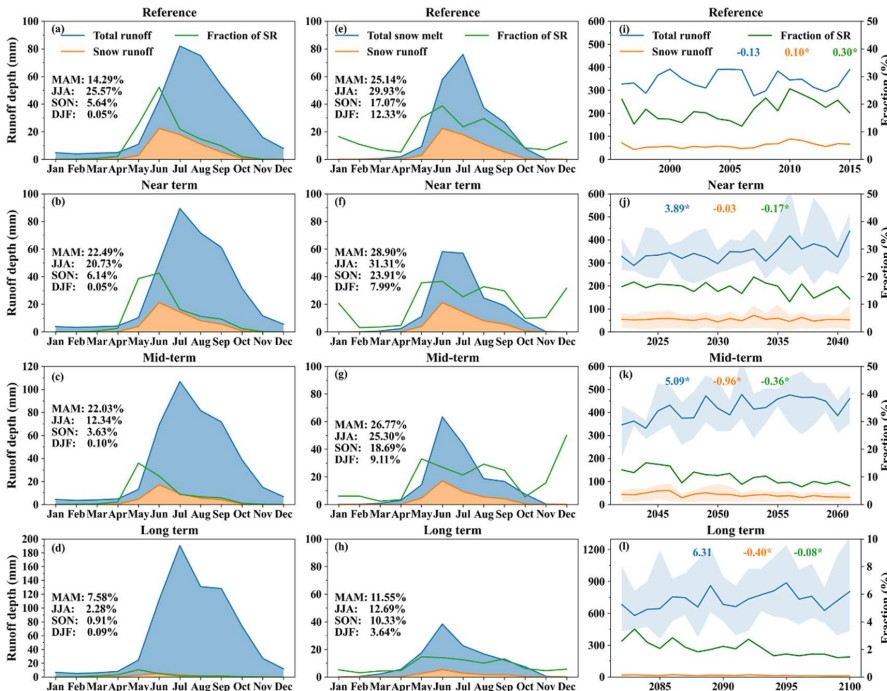

Figure. 15 Same as in Figure 14 but for SSP585.



The seasonal changes in the total runoff, snowmelt runoff, and snowmelt during all of the periods under the different SSPs were analyzed further. The intra-annual changes in the total runoff would be very similar to those of the precipitation, and the hydrograph would remain largely consistent in all periods under SSP126 and SSP585, with 60% of the annual total runoff occurring in summer and the peak flow occurring in July (Figs. 14a–d and 15a–d). There would be a very obvious change in the total snowmelt, that is, the peak snowmelt would shift from July to June after the reference period, but such a change would not occur in the snowmelt runoff (Figs. 14e–h and 15e–h). The snowmelt runoff would also be mainly concentrated in summer, accounting for about 85% of the annual snowmelt runoff during the reference period, and it would sharply increase in May, peak in June, and gradually decreased until it almost reaches zero in November. However, this hydrograph pattern would shift in the mid-term and long term under SSP585 due to the large decrease in snowmelt runoff in summer and the peak flow would be greatly reduced (Figs. 14a–d and 15a–d). Compared to the reference period, there would be a consistent decrease in the spring and winter total runoff in all of the periods under SSP126 and SSP585, except in the long term under SSP585 (Table 2). However, the total runoff would increase in summer and autumn in all of the periods under all of the SSPs, and it would increase to about 433 mm (217 mm) and 229 mm (106 mm) in summer and autumn, respectively, under SSP585 (SSP126) by the end of the century. The snowmelt would consistently increase in winter and spring in all of the periods under SSP126 and SSP585, and it would markedly decrease in summer and autumn in all of the periods. Similarly, the snowmelt runoff in spring is projected to increase in all of the periods under all of the SSPs, and it would slightly increase in winter and decrease in summer and autumn in all of the periods, except in the near term and long term under SSP126.

Table S6 shows the contributions of the multi-year average and seasonal snowmelt runoff to the annual total runoff in all of the periods under SSP126 and SSP585. The contribution of the snowmelt runoff to the annual total runoff (SR/TR) was about 17.6% in the reference period, and it would exhibit increasing–decreasing–increasing trends


from the near term to the long term under SSP126. The SR/TR is projected to account for about 19% of the total runoff by the end of the century. However, the SR/TR would constantly decrease from the near term to the long term under SSP585, and it would

account for only about 2% of the total runoff by the end of this century. Compared to the reference period, the SR/TR in spring is projected to increase in all of the periods under all of the SSPs, except in the long term under SSP585, and the snowmelt runoff in summer would contribute the most to the annual total runoff. Figures 10a–d and 11a–d show that on the monthly scale, the SR/TR was the largest in June during the reference

period. However, it would gradually shift to May from the near to long term under SSP585, and it would shift back to June in the long term under SSP126. Moreover, the maximum SR/TR occurred in summer in the reference period, but it would shift to spring after the mid-term under all of the SSPs due to climate warming. Regarding the total snowmelt, the contribution of the snowmelt runoff to the total melt (SR/TM) would

exhibit a bimodal pattern. The first peak was in June during the reference period, but it would shift to May under climate warming. The other peak was in August. In short, the above results indicate that the changes in the snowmelt under all of the SSPs would not be the primary reason for the increase in the annual total runoff, but its contribution cannot be ignored.





Table 3. Changes in the seasonal total runoff, snowmelt, and snowmelt runoff in the US basin under SSP126 and SSP585 compared to the reference period (1995–2014).

| Variable | SSP | Period | Spring | Summer | Autumn | Winter |
|---|---|---|---|---|---|---|
| **Total runoff (mm)** | SSP126 | Near term | 15.7 | 214.4 | 106.1 | 11.6 |
| | | Mid-term | 17.8 | 232.4 | 113.2 | 13.1 |
| | | Long term | 18.4 | 216.5 | 106.3 | 12.9 |
| | SSP585 | Near term | 18.1 | 210.7 | 105.3 | 12.6 |
| | | Mid-term | 19.9 | 257.6 | 125.2 | 14.5 |
| | | Long term | 38.5 | 433.0 | 228.8 | 23.6 |
| | Reference | | 20.1 | 199.8 | 104.7 | 16.4 |
| **Total snowmelt (mm)** | SSP126 | Near term | 13.1 | 144.2 | 28.6 | 0.1 |
| | | Mid-term | 14.5 | 134.0 | 27.2 | 0.1 |
| | | Long term | 13.4 | 139.1 | 27.2 | 0.1 |
| | SSP585 | Near term | 14.1 | 139.6 | 26.8 | 0.1 |
| | | Mid-term | 17.8 | 125.7 | 24.3 | 0.2 |
| | | Long term | 25.2 | 77.7 | 20.1 | 0.6 |
| | Reference | | 11.4 | 170.7 | 34.6 | 0.1 |
| **Snow runoff (mm)** | SSP126 | Near term | 4.1 | 53.9 | 8.7 | 0.0 |
| | | Mid-term | 4.3 | 37.9 | 5.5 | 0.0 |
| | | Long term | 4.7 | 55.4 | 8.7 | 0.0 |
| | SSP585 | Near term | 4.1 | 43.7 | 6.4 | 0.0 |
| | | Mid-term | 4.8 | 31.8 | 4.5 | 0.0 |
| | | Long term | 2.9 | 9.8 | 2.0 | 0.0 |
| | Reference | | 2.9 | 51.1 | 5.9 | 0.0 |

## 5. Discussion

### 5.1 Impacts of snow changes on runoff

According to the climate projections of the two SSPs, the future climate of the US

basin would become warmer and wetter, which would lead to a continuous reduction in the snowfall and accelerated melting of the SC during 2021–2100. In addition, the reduction of the annual snowfall and SC would be greater under SSP585 than under SSP126. Moreover, it was found that there are inconsistencies in the seasonal fluctuations of the simulated result in the different periods under SSP126 and SSP585.

For example, the summer and autumn SC would increase in all of the periods under





SSP126 compared to the reference period, but it would significantly decrease under SSP585. Moreover, a snow-free summer would likely occur in the long term under SSP585. The above phenomena were also reflected in the snow storage, snowmelt, and snowmelt runoff. This is mainly due to the differences in the GCM models, that is, the

warming rate of SSP58 is stronger than that of SSP126 during 1995–2100 (Fig. 8). Although SSP126 includes warming in the different periods, it has very stable changes in temperature and precipitation. Whereas SSP585 is accompanied by strong warming and significant precipitation from the near term to the long term, which prompts the shift from more solid precipitation to liquid precipitation in each season, resulting in

dramatic decreases in the snowfall, SC, and snow storage (Figs. 9, 10, and Table 1).

In addition, there would be significant differences in the response of the SC to climate warming in the different elevation ranges, and it was found that the SC in the low elevation regions (< 4500 m) would be more susceptible to climate warming than that at mid-high elevations. This is because the temperature and precipitation variations

are mainly dependent on an elevation gradient. Under the same temperature increase, the warming rate would be faster in the low elevation areas than in the mid-high elevation areas, particularly in areas below 4000 m a.s.l. on the TP. This is more likely to cause a large amount of snowfall to be converted into rainfall and increase the melting, which would result in a significant decrease in the amount of snowfall (Hock et al.,

2019; Marty et al., 2017; Kapnick et al., 2014), especially in spring when more snowfall would occur. Moreover, the decrease in the SC in the low elevation areas would also increase the absorption of solar radiation by the surface, increasing the surface temperature and further melting of the snow (Scherrer et al., 2012). However, the response to the increase in temperature would be slower in the high elevation areas than

in the low-elevation areas, which would result in more precipitation in the high elevation areas due to the effect of the terrain uplift, so there would be a correspondingly greater amount of snowfall (Hock et al., 2019). This change pattern would remain at lower elevations until the end of the century. However, in the long term, under SSP585, the SC in the high elevation areas would also be greatly reduced under the effect of the





large increase in temperature, and the ablation time of the SC would also be 1–2 months

earlier than during the reference period. This is because the warming rate would slow

down by the end of the century under SSP126, while the sharp temperature increase

would be more intense in the long term under SSP585 (Fig. 8 and Table S1).

        For both SSPs, the increase in the temperature would lead to a reduction in the

snow storage over the projected period, leading to a continued reduction of the total

snowmelt, and further causing a reduction in the snowmelt runoff, particularly in the

cold season (Figs.13, S4, and S5). We found that in the projected period, the spring and

winter snowmelt and snowmelt runoff would increase compared to the reference period,

especially in May (Fig. S6), Moreover, the reduction in the spring snowfall would lead

to a reduction in the amount of snowpack that can be stored in the spring. These factors

are the reasons why the meltwater and runoff in the summer would be greatly reduced

during the projected periods. However, the increased snowmelt in spring would cause

an increase in the snowmelt runoff, which would make up for the reduction in the total

runoff in spring to a certain extent and would play an important role in alleviating the

drought before the monsoon period (Table S7 and S8). Although the precipitation would

increase in autumn, the amount of snowfall would be very limited due to the influence

of the increase in temperature, which would result in a decrease in the snowmelt and

snowmelt runoff in autumn during the projected period (a significant decrease under

SSP585, and a weak balance under SSP126, Table S8, and Fig. S6). Hock et al. (2019)

and Nepal et al. (2021) pointed out a similar pattern of snowpack meltwater in alpine

regions, and they attributed this pattern to increased rainfall due to the increase in

temperature. During the projected period, the peak snowmelt in the US would shift from

July to June, but the pattern of the total runoff would differ from that of the snowmelt.

The total runoff would still peak in July (Figs. 14a–d and 15a-d), which is mainly

influenced by the monsoon precipitation (Figs. 9 and 10c–d). Thus, the annual

hydrological curve would remain unchanged in the future. This finding is consistent

with that of Su et al. (2016), that is, the runoff patterns of rivers in monsoon regions

would remain stable in the future. We also found that the proportion of snowmelt runoff





to snowmelt would continue to decrease. In addition to climate warming, another
possible explanation for this is that most of the snowmelt infiltrates into the soil and is
stored in the snowpack, and it a lot can also evaporate during runoff according to the
calculations of the WEB-DHM-sf model. Under future climate warming and the
continuous increase in precipitation, in view of the very small glacier area in the basin
(< 3%), the influence of the snow on the hydrological processes in the basin would
weaken, and precipitation would be the dominant factor affecting the runoff changes in
the future (Figs. 10c, d).

## 5.2 Comparison with other studies

Although there are some differences in the driving data, models, and periods used
in previous studies, the overall trends and patterns of the snow variables and runoff in
the US are still comparable and consistent. In this study, it was found that the total
runoff exhibited an insignificant decreasing trend ($-0.13$ mm yr$^{-1}$) during the reference
period, which is consistent with the recent findings of Yang et al. (2021). However,
there are inconsistencies with several previous studies based on site observations that
have reported a decrease in the total runoff, which is mainly due to the difference in the
study periods. For example, our results also show that the total runoff increased during
1981–2018 (Figure S7). The total runoff would also increase, which is consistent with
the results of Lutz et al. (2014), Su et al. (2016), Zhao et al. (2019), and Khanal et al.
(2021) for the projected period. A similar distribution pattern and seasonal changes in
the SC were reported by Kraaijenbrink et al. (2021), who also found that the SC would
decrease the most at low elevations, in summer and winter. They argued that the shallow
SC at the edge is more susceptible to warming, and the rapid reduction in the snowpack
would lead to a shorter snow season, which would also cause a decrease in the albedo
and would eventually form an elevation-dependent warming feedback cycle. The
interannual and seasonal changes in the future snowmelt runoff are also very similar to
the predictions of previous studies (Su et al., 2016; Zhao et al., 2019; Khanal et al.,
2021; Kraaijenbrink et al., 2021). All of these studies found that the snowmelt would
decrease and advance to spring, although there may be differences in a certain period.


This can be mainly explained by the shift in the snow-rain and the increased snow melting rate due to warming, as well as the decrease in the snowfall in the US. The contribution of the snowmelt runoff to the total runoff during the reference period was about 17.6%, which is between the values of 13.4% (Yang et al., 2021) and 28.3% (Lutz et al., 2014) reported in previous studies. This is largely due to the differences in the forcing data and hydrological model used to describe the physical snow-related processes. The contribution of the snowmelt runoff to the total runoff under SSP585 is consistent with previous studies that predicted significant decreases (Su et al., 2016; Zhao et al., 2019; Khanal et al., 2021). We also concluded that the snowmelt runoff would be important but would not be the main factor controlling the changes in the total runoff in the basin.

### 5.3 Uncertainties and limitations

The main uncertainties in this study were as follows. First, very limited observed runoff data (for only 6 years) were used in the calibration and validation of the model. The lack of long-term runoff observations may cause uncertainties in the simulation results for the other periods. Second, although the ERA5 precipitation data were better than the other products in terms of the temporal and spatial distributions, most studies have pointed out that this product overestimates the precipitation during the monsoon period in the high mountains (Yang et al., 2021; Khanal et al., 2021). The ERA5 product was evaluated at a single point, and the historical forcing data were not corrected. Although this effect could be reduced through model parameter calibration, it would inevitably lead to the overestimation of variables such as snow cover and runoff. For example, Khanal et al. (2021) pointed out that the cold bias in the ERA5 (even after bias correction) is the main cause of the overestimation of the winter snow cover, but an insufficient description of the snow sublimation and other physical mechanisms in the model may also lead to excessive snow cover. In addition, although the climate model data selected in the study can reflect the uncertainty range of all of the GCM datasets, there is still a great deal of uncertainty at the basin scale. Moreover, based on the ERA5, a simple bias correction method was used to conduct a secondary correction





of the GCM, which only ensured the consistency of the relative change trend. However, it did not improve the accuracy of the future predictions of the meteorological variables, such as the precipitation and temperature frequency distribution and seasonal variations,

805 which may cause some uncertainty in the simulation results (Khanal et al., 2021; Zhao et al., 2019; Su et al., 2016). Therefore, to improve our understanding of the effects of the synergistic changes in the cryospheric components under climate change on the hydrological processes in high mountain areas, it is necessary to strengthen the construction of the plateau observation network, collect long term observations, and

810 improve the data quality. In addition, it is necessary to enhance our understanding of the related hydrological processes through in situ experiments and to calibrate more reliable parameters to further improve the physical processes in the model. Future research needs to use more reliable bias correction methods and downscaling methods to improve future predictions. In addition, the SSP245 scenario, which is closer to the

815 current state of development, was not considered here because of the data limitations.

## 6. Conclusions

In this study, we used multi-source reanalysis data and four GCMs under two different SSPs to drive the validated cryosphere–hydrological model (WEB-DHM-sf) to analyze the impact of future snow changes on runoff in the US basin. The main

820 conclusions of this study were as follows:

1) From 1995 to 2100, the annual average temperature in the US was projected to significantly increase under SSP126 (0.2°C 10 yr$^{-1}$) and SSP585 (0.7°C 10yr$^{-1}$). The annual precipitation would also significantly increase under SSP126 (5 mm 10yr$^{-1}$) and SSP585 (27.8 mm 10yr$^{-1}$). The climate of the US would become warmer and

825 wetter in the future. On the seasonal scale, the temperature would significantly increase in all of the seasons, while the precipitation would increase in summer and autumn and decrease in the winter and spring. In the future, the US was projected to become warmer and wetter during the monsoon season and warmer and drier during the non-monsoon season.





2) From the perspective of the entire study period, the annual and seasonal snowfall in the basin would significantly decrease, i.e., by 13–16% in the near term and 16–49% in the long term. Overall, the snowfall would also decrease in all of the seasons, with the greatest decrease occurring in summer and autumn, i.e., by 44–85% and 21–59% by the end of the century, respectively. The decrease in snowfall would directly affect the changes in the snow cover. The snow cover in the low elevation areas would decrease significantly due to the warming, especially in winter and spring. Melting would also accelerate in the middle-high-elevation areas due to the increase in temperature under SSP585, especially in the long-term, and the ablation of snow would shift from May–March. The massive continued snow melting would further reduce the snow storage. Compared with the reference period, the snow storage in the US in the future would exhibit negative growth in terms of both annual average (decreased by > 30% under SSP126 and by 85% under SSP585) and seasonal (decrease by > 30% in summer and autumn under SSP126 and by 90% under SSP585) changes. Climate warming would lead to unfavorable development of snow in the basin.

3) During the reference period, the contribution of the snowmelt runoff to the total runoff was about 17.6%. Under the large temperature increase scenario, this contribution would become continuously smaller in the future and be about 2% by the end of the century, leading to changes in the total runoff from a snow-rain-dominated pattern to a rain-dominated pattern. Under the SSPs, the snow-rain runoff pattern would be maintained. The annual total runoff in the US would increase significantly in the future, which would also increase the availability of water resources in the basin. At the seasonal scale, the total runoff would decrease in winter and spring and increase in summer and autumn, but the total annual hydrograph would remain unchanged. The increasing peak runoff may increase the risk of flooding in the future. The snowmelt runoff would significantly decrease, except under SSP126, which would result in a weak balance. This showed that the meltwater peak would advance to June, and the largest proportion would occur in





May. The increase in the spring snowmelt would make up for the reduction in the spring total runoff caused by the reduction in rainfall, thereby ensuring the availability of water resources in the basin during the growing season in the spring and alleviating the spring drought to a certain extent. The advance in the snowmelt would also lead to the decline of snowmelt in summer and changed the pattern of snowmelt runoff in summer. These changes would have some impacts on the availability of water resources, ecosystem, and agriculture in high mountain regions. As such, to reduce the risk of summer rain floods and spring droughts, it would be necessary to actively adjust the water resource allocation scheme to adapt to the impacts of climate change.

## Data availability

All data used in this paper from following sources: CMFD and the second glacier inventory dataset of China (version 1.0) (http://data.tpdc.ac.cn/), ERA5 (https://cds.climate.copernicus.eu/), GLDAS, MERRA2, and MODIS LST (NASA, https://search.earthdata.nasa.gov/), MSWEP (http://www.gloh2o.org), ISIMIP 3b data (https://data.isimip.org/), DEM (SRTM, https://srtm.csi.cgiar.org/), land use maps (USGS, http://edc2.usgs.gov/glcc/glcc.php), soil data (FAO, http://www.fao.org/geonetwork/srv/en/main.home), LAI and FPAR (GLASS, http://www.geodata.cn/thematicView/GLASS.html), Meteorological observation data (CMA, http://data.cma.cn/), MOYDGL06* (PANGAEA, https://doi.pangaea.de/10.1594/PANGAEA.901821).

## Author contribution

Chenhao Chai and Lei Wang designed the research and text organization; Chenhao Chai analyzed data and wrote the manuscript; Jing Zhou, Hu Liu and Jingtian Zhang provided technical guidance. Lei Wang, Deliang Chen, Jing Zhou, Yuanwei Wang, and Tao Chen provided suggestions and reviewed the manuscript.

## Competing interests

The authors declare that they have no conflict of interest.





## Acknowledgements

This research was supported by the National Natural Science Foundation of China (92047301 and 41988101), the Strategic Priority Research Program of the Chinese Academy of Sciences (XDA19070301), and the Second Tibetan Plateau Scientific Expedition and Research Program (STEP) (2019QZKK020604). We sincerely thank Prof. Beck, who generously provided the precipitation datasets of MSWEP 2.8.

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
