# Peer review of "Future snow changes and their impact on the upstream runoff in"

_Hydrology and Earth System Sciences, 2022_

## Author Comment (AC1)

**Reply to Referee #1 Comments**

**General Comments:**

The current study is well thought out, while being presented and organized in a clear manner. The results are important for the scientific community and beyond (water managers and water users within the basin), and this study could play a role in initially quantifying of the types of changes that may occur in the future within the context of a warming climate. I would recommend the manuscript be published, after some minor concerns are addressed. The paper suffers from a lack of validation datasets at the relevant spatial and temporal scales. This is often the case with remote, data sparse, mountainous environments that are the subject of long-term snow research questions. The authors seem to creatively address those problems, but need to make some of the limitations of the validation datasets more clear. I will divide my comments into two categories: minor concerns and minor suggestions.

**Reply:** Many thanks for the positive comments and suggestions. For the limitations of the validation datasets used in this study, we will elaborate and supplement in detail.

**Minor Concerns:**

1. Line 148: This is the first time the SSP scenarios are mentioned. They are never defined and this is problematic for the study since it relies heavily on two of the SSP scenarios. Please briefly define the scenarios that are being used in this study and be clear about the differences between SSP126 and SSP585. This is important for readers that may be unfamiliar with these scenarios.

**Reply**: We have defined the SSP scenarios in the revised manuscript (Line 146–152): "SSP126: it is a combination of the low emission scenario RCP2.6 and the sustainable socioeconomic pathway SSP1, and it represents the low vulnerability, low mitigation pressure, and low radiative forcing (2.6 W m-2); SSP585: it is a combination of the high emission scenario RCP8.5 and the fossil-fueled development pathway SSP5, and it is the only shared socioeconomic pathway that can achieve an anthropogenic radiative forcing of 8.5 W m-2 by 2100, representing the worst development pathway."

2. Line 205: A limitation of this study is the model output resolution of 5km. In complex, mountainous terrain with high relief, a 5km grid cell introduces a huge amount of uncertainty and potential for inaccuracy, both for snow process modeling and for satellite snow remote sensing datasets. Can the authors address this scale

issue in the discussion and add some references to important snow scaling literature related to complex mountain environments? What do the authors think would change in the results if they could model at a significantly higher resolution?

**Reply**: We agree with the reviewer. The change of the model resolution will directly affect the terrain, that is, the slope, aspect, elevation, etc. will change accordingly, which will affect the model forcing variables that dependent on terrain. Following the suggestions, we further examined the performance of 3 km model resolution on the snow hydrological simulation based on the parameters of 5 km model resolution as follows (Lines 826-856):

"Here, the accuracy of snow hydrological simulation between the model resolution of 3 km and 5 km was compared. The NSE, $R^2$, KGE, RB and RMSE between the simulated and observed discharge at basin outlet during 1981-1987 based on the 3 km model resolution is 0.82, 0.8, 0.87, -7.93% and 346.12 $m^3 s^{-1}$, respectively (Fig. 8a). It shown that the accuracy of simulated discharge based on the 3 km resolution is slightly improved at basin outlet compared to that of the 5 km model resolution. Meanwhile, the 8d and monthly FSCA of former was slightly improved at basin-scale compared to that of the latter according to the $R^2$ that is improved from 0.73 (8d), 0.56 (monthly) to 0.74, 0.59, respectively (Figs. S9 b–c and Figs. 5 b–c). Overall, the improvement was not obvious between 3 km resolution and 5 km resolution, which may be caused by the following reasons: the model (WEB–DHM–sf) used in this study takes into account the sub–grid parameterizations, that is, a model grid consists of a set of geometrically symmetrical hillslopes that are the basic hydrological units (BHUs) of the model. The topography parameters (slope, length) of the hillslope in each model grid can be calculated by a fine DEM with 90m resolution (Wang et al., 2009a). The sub–grid parameterization scheme can enable more detailed characterization of basin topography information such as slope, aspect, sky view factor, elevation etc. (Etchevers et al., 2001), thereby retaining the important information of topography-induced spatial variation in forcing inputs (the solar radiation, albedo, precipitation, wind etc.), which would reduce simulation bias caused by model resolution (Winstral et al., 2014; Sohrabi et al., 2019). Furthermore, the original resolution of the forcing data input to the model was above 10 km. Downscaling from low resolution to high resolution (3 km and 5 km) through

interpolation can't improve the accuracy of these variables that affect the snow hydrological simulation, that is because coarse-scale data cannot transfer more effective information to the interpolated fine-scale data by downscaling scale (Sohrabi et al., 2019). However, this didn't mean that it is not important to improve the model resolution, but the model resolution suitable for a specific watershed should be selected according to the characteristics of the watershed, the snow hydrological model used, and the accuracy of the forcing data."

[Figure]

Figure. S8 Simulated and observed (a) 3 km and (b) 5km daily discharges at Jiayuqiao (JYQ) station from 1981 to 1987. The calibration and validation periods were 1981–1983 and 1984–1987, respectively.

[Figure]

Figure. S9 Comparison of the simulated 3km fraction of the snow cover area (FSCA) and the MOYDGL06* time series in the USR basin. (a) 8-day FSCA; (b) multi-year mean monthly FSCA; and (c) variations in the monthly FSCA during 2003–2018.

3. Line 218-219: What type of evaluation did the authors conduct for the historical time period? How was the evaluation assessed and what metrics were used? They do not mention their methods in this paragraph (or anywhere in the publication?) and how they came to the conclusion that ERA5 was the best.

**Reply**: We thank the reviewer for pointing this out. We have made corresponding revisions as follows (Line 217-234):

"To better evaluate the applicability of four reanalysis precipitation products (ERA5, GLDAS, MERRA2, MSWEP) in the USR, here we mainly used several statistical indices to evaluate the products based on the meteorological observation data, including Correlation coefficients (CC), Mean error and Root mean square error (RMSE). Moreover, the probability of detection (POD), false alarm ratio (FAR), missing alarm rate (MAR), and Critical success index (CSI) were calculated to check the capture capability of rainfall events from reanalysis products (Ebert et al., 2007; Tian et al., 2009). It can be seen from Figs. S1 and S2 that, except for GLDAS, the statistical indicators of other precipitation products are relatively consistent, but if the international exchange station is excluded (Naqu and Dingqing), ERA5 and MSWEP have good performance (CC is about 0.5). In addition, ERA5 has higher POD (average

about 0.97) and lower FAR (average about 0.44) as a whole, which is also shown on CSI (average about 0.56), implying that ERA5 has better characterization of rainfall events."

**Supplemental Figure**

[Figure]

Fig. S1 Comparison of the daily precipitation values between the gauge observation and different operational global products during 2000-2019. The results of ERA5, GLDAS, MERRA2, and MSWEP2.8 are given from the first to the fourth column.

[Figure]

Fig. S2 Evaluation of capturing ability of different global operational products to precipitation events. Effective precipitation is defined as daily precipitation greater than or equal to 0.1 mm. (a) POD (probability of detection); (b) FAR (false alarm ratio); (c) MAR (missing alarm rate), and (d) CSI (critical success index).

4. Line 235: The delta method is mentioned here (and later in the publication) but never described. Please briefly describe this method for readers who are unfamiliar.

**Reply**: We have supplemented the delta method as follows (Line 247):

"The delta method (it is a simple linear bias correction method, see bias correction methods in Supplementary Material for details).

Supplemental Methodology:

Bias correction methodology for future scenario meteorological data

Using the delta method to correct the future monthly data of GCM needs to calculate the monthly correction factor that is the differences and fraction between monthly mean of historical observations and the monthly mean of GCM historical simulations (1995-2014). The correction factor is then multiplied or added to the future simulated data of the GCM for the corresponding month (Gleick et al., 1986; Hay et al., 2000). The calculation formula is as follows,

$$T_{fut,cor} = T_{fut,GCM} + \left( \bar{T}_{his,obs} - \bar{T}_{his,GCM,} \right) \tag{1}$$

$$P_{fut,\text{cor}} = P_{fut,GCM} \times \left( \frac{\bar{p}_{his,obs}}{\bar{P}_{his,GCM}} \right) \tag{2}$$

Where $T_{fut,cor}$ is the bias-corrected GCM future air temperature, near-surface air pressure, long-wave radiation and short-wave radiation for the 2021-2100; $P_{fut,cor}$ is the bias-corrected GCM future precipitation, wind speed, and specific humidity for the 2021–2100; $\bar{T}_{his,obs}$ and $\bar{p}_{his,obs}$ is the monthly mean of the observation of historical period (1995–2014); $\bar{T}_{his,GCM}$ and $\bar{P}_{his,GCM}$ is the monthly mean of the GCM simulation of historical period (1995–2014)."

5. Line 303: I've never seen a remotely sensed satellite temperature dataset used for validation in a snow modeling study. This does not mean using this dataset is invalid and please correct me if I'm wrong (and add some publication citations for clarity). The need for this validation dataset is likely due to the lack of other spatially extensive snowpack observational datasets, however it still seems a bit odd to me. I'm left wondering how this validation method/dataset performs at elevations in the Upper Salween River watershed where temperatures are most likely to influence snow processes? Also, how does this validation method/dataset perform during the months of the year that snowpack is accumulating and melting, rather than just looking over the entire 18-year time period?

Reply: LST is a key parameter in land-atmosphere interactions because it is the main factor controlling the surface-atmosphere sensible and latent heat fluxes (Wang et al., 2009). Assessment of the LST can be used to evaluate model performance in representing basin-wide energy processes. Due to the lack of heat flux observations, there is no energy-related analysis in this region. However, Satellite remote sensing offered the most feasible, consistent, and accurate means of global fields of land surface parameters (Sellers et al., 1997). In recent years, Moderate Resolution Imaging Spectroradiometers (MODIS) datasets with global coverage and high resolution, were widely used for model evaluations in geophysical studies (Sheng et al., 2009; Shrestha

et al., 2012; Corbariet al., 2014; Zhong et al., 2020; Zhang et al., 2020). The MOD11A2 has shown good agreement with the ground-based LST measurements on the western TP with mean difference of 0.27 K (Wang et al., 2007), and it provides us with a viable option to improve water and energy budget studies at the watershed scale. Therefore, we use MODIS LST to evaluate the performance of the model in the characterization of energy processes. This evaluation method is widely used in high mountain regions, such as Yangtze River (Qi et al., 2019), Nam Co Lake (Zhong et al., 2020), Selin Co Lake (Zhou et al., 2015), etc.

Following the reviewer's suggestion, we further verified the basin-averaged LST from the different elevations (Fig. AC2), and those in snow accumulation and ablation periods (Fig. AC1) during 2001-2018. From the point of view of snow accumulation and ablation period, the simulated LST during the snow accumulation period is better than that in the ablation period compared to MODIS LST. The daytime and nighttime LST simulated during the snow accumulation period are in good agreement with the MODIS LST ($R^2 > 0.7$), but the simulated LST in the daytime is generally lower (MB = -3.87K) and that in the nighttime is slightly higher (MB = 1.64K). While the accuracy of simulated LST has a difference between daytime and nighttime during the snow ablation period, the simulated daytime LST don't reproduce the MODIS daytime LST well ($R^2 = 0.21$, MB = -3.93K) compared to that in the nighttime ($R^2 = 0.67$, MB = 1.01K).

The simulated daytime and nighttime LST can generally better characterize the variation trend of the LST at different elevations in the USR basin, and also have good consistency. The simulated daytime and nighttime LST in high-elevation areas (> 5000m) are better than those in low-elevation areas (< 4500 m) based on the determining coefficient and mean bias. Moreover, the simulated nighttime LST at each elevation is better than those in daytime compared with MODIS LST. The simulated daytime LST at each elevation and nighttime LST at low elevation are lower than MODIS LST, while the nighttime LST at mid-high elevations is slightly higher.

We find that the accuracy of the simulated daytime LST is generally lower than that of the simulated nighttime LST regardless of that perspective. Overestimation for

the daytime was greater than the nighttime mainly due to the complex interactions of the surface energy balance during daytime. The LST is controlled by solar radiation absorbed by the surface canopy and the ground during daytime, and these factors lead to greater uncertainty in the simulated daytime LST in the USR basin with complex terrain, while the solar radiation can be ignored at nighttime, and the LST is mainly affected by the downward long wave radiation (Xue et al., 2013; Zhou et al., 2015).

[Figure]

Figure. AC1 Validation of simulated 8-day daytime and nighttime land surface temperatures in accumulation (a, b) and ablation periods (c, d) compared to MODIS 8-day satellite observations, averaged over the USR basin during 2001–2018.

[Figure]

Figure. AC2 Change in the 8-day simulated land surface temperature (day and night) in the different elevation bands compared to MODIS 8-day observations, averaged over the USR basin during 2001–2018.

**Minor Suggestions:**

1. The title is awkward and a little unclear. I would suggest something more along the lines of "Hydrological changes to runoff in the Upper Salween River from forecast changes to snowpack under climate warming scenarios". This is just a suggestion.

**Reply**: The main objective of this study is to simulate the changes in the snow-related hydrological processes in the USR. We used the WEB-DHM-sf driven by global climate model (GCM) in CMIP6 to predict the changes in the snowfall, snow cover, snow water equivalent (SWE), total snowmelt, snowmelt runoff during different periods (near-term: 2021-2040; Mid-term: 2041-2060; Long-term: 2081-2100) under different shared socioeconomic pathway scenarios (SSP126 and SSP585), and we further analyzed the impact of the snow changes on the runoff. Therefore, although the reviewer gives a good suggestion, the original title of the manuscript may be more in line with the major contents of the study.

2. Line 19: The authors chose to abbreviate Upper Salween to US but this may be confusing for readers based in the United States, which is also often abbreviated to US. Consider USR for Upper Salween River using instead of US?

**Reply**: Following the reviewer's suggestion, we have changed the abbreviation of Upper Salween River to "USR" in the revised manuscript.

3. Line 22: I would encourage the authors not to assume that everyone reading the abstract knows what these abbreviations (SSP126 and SSP585) mean without context or the full spelling

**Reply**: We have supplemented the abbreviations as follows (Lines 18–20):

"In this study, we aimed to project future snow changes and their impacts on the hydrology in the upstream region of Salween (USR) under two shared socioeconomic pathway scenarios (SSP126 and SSP585) using a physically-based cryosphere–hydrology model."

4. Line 51: I'm unsure what 'all walks of life' means in this context.

**Reply**: For this unclear expression, we have revised it as follows (Line 51):

"Accordingly, it not only has a strong effect on the regional hydrological cycle, but also provides abundant water resources for the industry, agriculture and residents in basins and supports about one-sixth of the world's population."

5. Line 84: The sentence beginning on this line is long and complex, consider splitting apart for clarity.

**Reply**: We shall do this as follows (Line 84):

"Previous studies have assessed several snow variables (e.g., snowfall, snow storage, SC, and snowmelt) and the hydrological processes related to snow under climate change on the TP based on in-situ observations and land surface snow/hydrological models. However, the impact of future snow changes on runoff is still unclear due to a lack of reliable data."

6. Line 99-100: Consider 'study area' instead of 'research object.'

**Reply**: We have made revisions as follows (Line 95–97):

"To better understand the effect of snow changes on the TP on runoff under climate change, in this study, the upstream region of Salween (USR) was selected as the study area."

7. Line 101: This is vague, I am not sure what the authors mean by 'complex underlying surface.' Same with line 105 'underlying surface.'

**Reply**: The meaning of 'complex underlying surface' mentioned in the text is the same as 'the underlying surface', both refer to the diverse and complex surface environment in the basin. In order to avoid ambiguity, we unified the two expressions, using 'complex underlying surface'.

8. Line 144: Use 'observational' instead of 'observation'

Reply: Revised.

9. Line 322: Are there any stats that can quantify the 'very close' and 'slightly overestimated' claims here, thus bolstering your argument?

**Reply**: The 'very close' and 'slightly overestimated' mentioned here both refer to the comparison between the simulated LST and the MODIS LST during the daytime and nighttime at the basin-average values. We can see the basin-average value of different seasons LST from Figure 4. We have revised it as follows (Line 336):

"The basin-average value of simulated seasonal LST during the nighttime was closer to the MODIS LST (DJF: MB = -2.25 K; MB = JJA: -1.07 K; MAM: MB = -1.5 K; SON: MB = -0.66 K) than that in daytime, and the simulated seasonal LST during the daytime was underestimated in winter (MB = -5.38 K), spring (MB = -7.24 K), and autumn (MB = -2.76 K) and was slightly overestimated in summer (MB = 0.86 K)."

10. Line 365, Figure 6: Using the off-white color to represent no snow and the blue color to represent snow is not easy to interpret here. Anytime white is used in a snow study it is often interpreted as the presence of snow.

**Reply**: We have made revisions as follows (Line 385):

[Figure]

Figure. 6 Comparison of the MODIS 8-day snow cover product (MOYDGL06*) and the simulated 8-day snow cover in the USR basin in 2017. The time interval is 16 days

11. Line 415, Figure 8: These figures are excellent and easy to interpret. However, the SSP585 line becomes difficult to see during the long-term portion of the figure in both (a) and (b). Consider changing the color of the trend line to clarify.

**Reply**: We have made changes as follows (Line 437):

[Figure]

Figure. 8 Relative changes in the annual precipitation and mean annual temperature from 1995 to 2100. The black, bule, and green lines represent the precipitation and temperature during the reference period (1995–2014) and under CMIP6 SSP585 (2015–2100) and SSP126 (2015–2100), respectively. The rectangular shaded areas are the near-term (pink: 2021–2040), mid-term (yellow: 2041–2060), and long-term (red: 2081–2100) periods. The shading around the lines represents the fluctuation range of the data, and the upper and lower ranges are 95% and 5%.

12. Line 436, Figure 9: Again, excellent and intuitive figure. The only thing that needs clarification is the dashed vertical lines, I'm not sure what they represent.

**Reply**: The dotted line is just a hint that means that the precipitation in April and October during future period (near term, mid-term, long term) is higher than that in the reference period (1995-2014).

Line 457:

[Figure]

Figure. 9 Seasonal changes in the temperature and precipitation in the reference period, the near term, the mid-term, and the long term under (a, c) SSP126 and (b, d) SSP585. The shading around the lines represents the fluctuation range of the data, and the upper and lower ranges are 95% and 5%. The dotted line of figure b and d means that the precipitation in April and October during future periods higher than that in the reference period.

13. Line 474, Figure 10: In figure 10 (c) and (d) does the solid line represent snowfall? If so, that is not clear from the legend.

**Reply**: This solid line does represent snow, and we have modified the legend for this figure.

Line 498:

[Figure]

Figure. 10 Snowfall changes and the ratio of snowfall to total precipitation (TP). (a, b) Annual changes in and proportion of snowfall, (c, d) seasonal changes in snowfall and rainfall, and (e, f) the ratio of snowfall to total precipitation on the seasonal scales in the USR basin during different periods under SSP126 and SSP585 compared to the reference period. The shading around the lines represents the fluctuation range of the data, and the upper and lower ranges are 95% and 5%.

14. Line 479: I am not sure what is meant by 'fluctuation range of the data'.

**Reply**: The fluctuation range of the data indicates the spread in projections for the

future period when forced with the ensemble of 4 GCMs which can also be understood as the very likely range of simulations and the uncertainty range (IPCC, 2021). The shaded band around the line denotes the interquartile range, and the upper and lower ranges are 95% and 5%.

15. Line 497: I think FSAC should be FSCA here.

**Reply**: Revised.

16. Line 500: I think 'ere' needs to be deleted.

**Reply**: We have removed this misspelling in the revised manuscript.

17. Line 605 (Fig14) and line 626 (Fig15): These figures are very informative while also being a little complicated to understand. My suggestion is to: 1) remove the seasonal average percentage stats from the figures and put them in a table and 2) separate out the third column as a separate figure that is reported alongside the new table. Then, the authors could group the SSP126 runoff figs and the SSP585 runoff figs into the same figure. In other words, Fig14 column 1 & 2 would be beside Fig 15 column 1 & 2 in a new Figure14. Likewise, Figure14 column 3 would be beside Figure15 column 3 in a new Figure 15. I think this would help simplify the experience for the readers.

**Reply**: Following the reviewer's suggestions, we have removed the seasonal average percentage stats from the figures 14 and 15, and put them into a table in the supplement file (Tables S7 and S8). we have modified Figures 14 and 15 in the revised manuscript, specifically as follows,

**Supplementary information:**

Tab. S7 Change and proportion of future seasonal snow runoff to total runoff comparing to the reference period.

| SSP | Period | Spring(mm) | | | Summer(mm) | | | Autumn(mm) | | | Winter(mm) | | |
|-----|--------|------|-----|------|-------|------|------|-------|-----|-----|------|-------|------|
| | | TR | SR | % | TR | SR | % | TR | SR | % | TR | SR | % |
| SSP126 | Near term | 15.7 | 4.1 | 25.9 | 214.4 | 53.9 | 25.1 | 103.0 | 8.7 | 8.5 | 11.6 | 0.01 | 0.09 |
| | Mid-term | 17.8 | 4.3 | 24.2 | 232.4 | 37.9 | 16.3 | 113.2 | 5.5 | 4.9 | 13.1 | 0.01 | 0.08 |
| | Long term | 18.4 | 4.7 | 25.6 | 216.5 | 55.4 | 25.3 | 106.3 | 8.7 | 8.2 | 12.9 | 0.01 | 0.09 |
| SSP585 | Near term | 18.1 | 4.1 | 22.5 | 210.7 | 43.7 | 20.7 | 104.3 | 6.4 | 6.1 | 12.6 | 0.007 | 0.05 |
| | Mid-term | 21.7 | 4.8 | 22.0 | 257.6 | 31.8 | 12.3 | 125.2 | 4.5 | 3.6 | 14.5 | 0.01 | 0.1 |
| | Long term | 38.5 | 3.0 | 7.6 | 433.0 | 9.9 | 2.3 | 228.8 | 2.1 | 0.9 | 23.6 | 0.02 | 0.09 |

| Reference | 20.1 | 2.9 | 14.3 | 199.8 | 51.1 | 25.6 | 104.7 | 5.9 | 5.6 | 16.4 | 0.01 | 0.05 |

Tab. S8 Change and proportion of future seasonal snow runoff to total snowmelt (TM) comparing to the reference period.

[revised manuscript text omitted]

19. Line 746: I am unsure what 'a lot' means in this context, consider quantifying it or maybe remove?

**Reply**: We have removed it in the revised manuscript.

20. Line 766: I am confused by the term 'at the edge' here, please clarify.

**Reply**: The 'at the edge' indicates the lower elevation areas. The changes of snowpack at lower elevation regions are more strongly than that at mid-high elevation regions, since the effects of climate change on the shallow snowpacks at low elevations can be relatively immediate. Eventually, these reductions in snow may translate to shorter snow seasons and a rapid associated decrease in albedo, which contributes to elevation-dependent warming feedbacks. We replaced it in the revised manuscript with at low

elevation regions.

21. Line 839: I am unsure of what is meant by 'massive continued snow melting' here.

**Reply**: What we want to express here is that a large amount of snowpack is melting in the USR, which further lead to a continuous reduction in snow storage. We have revised it as below (Line 880):

"A large amount of snowpack is melting in the USR, which further lead to a continuous reduction in snow storage."

22. Line 857: I am unsure of what is meant by 'result in a weak balance' here.

**Reply**: We have made the following modifications (Line 897):

"The snowmelt runoff would significantly decrease, except the long term of the SSP126, and the meltwater peak would advance to June, with the largest proportion would occur in May."

---

## Author Comment (AC2)

**Reply to Referee #2 Comments**

In this study, Chai et al. simulated future snow changes and their impacts on the upstream runoff in Salween. This is an important study for water resources in future. Overall, this study explained well. However, there are still some questions needing to be clarified.

**Reply**: Many thanks for the positive comments. For the questions raised by the reviewer, we will elaborate and supplement in detail as follows.

1. The abbreviation of US easily confused readers as United states.

**Reply**: We have changed the abbreviation of the Upper Salween River to "USR" in the revised manuscript.

2. ERA5 precipitation was better than that of CMFD. How about other CMFD variables compared to ERA5?

**Reply**: Following the reviewer's suggestion, we further verified other variables of ERA5 and CMFD based on six meteorological stations. Taking into account the availability of data (ERA5 hourly data on single levels have only temperature, pressure, wind), we chose the variables of the air temperature, air pressure and wind speed for verification during 1995-2018. The result is shown from the Fig. AC1.

[Figure]

Fig.AC1 Comparison of the daily temperature, pressure, wind values between the CMA observation and different operational global products (CMFD, ERA5) during 1995-2018. The blue points are

ERA5-Obs, and the yellow points are the CMFD-Obs.

For the air temperature, the CMFD is closer CMA observation than the ERA5, its $R^2$, MB and RMSE are 0.96, -1.09 °C and 1.93 °C, respectively. For the air pressure, the $R^2$ MB and RMSE between the CMFD and observation data is 0.81, -10.66 hpa and 16.69 hpa, which is better than that of the ERA5 (0.25, -40.82 hpa, 47.87 hpa). For the wind, the CMFD have a higher $R^2$ (0.68), a lower MB (0.1 m s$^{-1}$) and RMSE (0.75 m s$^{-1}$) than that of the ERA5 (0.22, -0.63 m s$^{-1}$, 1.37 m s$^{-1}$) compared to the CMA observation. Overall, the CMFD dataset performed much better than ERA5.

3. How about the consistency using variables from different dataset to force the model?
**Reply**: The model input data were based on the CMFD dataset (precipitation, air temperature, air pressure, wind speed, specific humidity, as well as downward shortwave and longwave radiations). After that, we replaced the CMFD precipitation with other precipitation products (ERA5, GLDAS, MERRA2, MSWEP) and verified the simulation accuracy of different precipitation products by the basin outlet. To ensure the consistency of spatiotemporal resolution, all variables were resampled to 3 hours temporal resolution and interpolated to 5km spatial resolution. We finally found that the optimal combination of the ERA5 precipitation and the other CMFD meteorological variables performed very well in the USR basin (Fig. AC 2).

[Figure]

Figure. AC2 Simulated and observed (a) daily and (b) monthly discharges at Jiayuqiao (JYQ) station from 1981 to 1987. The calibration and validation periods were 1981–1983 and 1984–1987, respectively.

4. There were too many kinds of data in section 3.2. Suggest to give subtitles to make them clear.

**Reply**: We did this in the revised manuscript.

5. The discharge was partly the result of the snow change. Why was the discharge evaluated before snow and temperature?

**Reply**: First, we calibrated and validated the soil hydraulic parameters of the model through the observed discharges. After the model parameters determined, we verify the process-based variables of the model (e.g., the snow by FSCA, and the temperature by LST).

6. The LST RMSE between Modis and the simulation was high as 6.11 K in the day. The bias and RMSE in winter night was higher than that shown in Figure 3b. Is the precision of the simulation acceptable? Could you improve the simulation? As you said, the difference was caused by CMFD data. How about results using other forcing data, such as ERA5 that has better precipitation data than CMFD?

**Reply**: Although the RMSE and bias between Modis and the simulation was not perfect, the simulated value could better reflect the variation trend of LST during the daytime

$(R^2 = 0.69)$ and the nighttime $(R^2 = 0.91)$. Following the reviewer's suggestion, we used the GLDAS temperature (Tair) to drive the model. The result was shown in Fig. AC3, the simulated LST by the GLDAS Tair also shown that the simulation results of nighttime are better than that in the daytime, and the simulation results of LST are not improved comparing to the CMFD Tair. As we can see from the Fig. AC3 and Fig. 3, the accuracy of Tair has a great impact on the simulated daytime LST $(CC \geq 0.85)$, but there is a bias between GLDAS Tair and observed Tair (Fig. AC4). In addition, factors such as complex terrain and cloud coverage in the USR basin may also cause misjudgment of the MODIS LST, which also may lead to bias in the 8-day MODIS LST product.

[Figure]

Figure. AC3 Comparison of 8-daily LSTs between model simulations (simulated) and MODIS observations (observed) during daytime (upper) and nighttime (lower) averaged for the URS from 2001 to 2018. Here, the input air temperature (Tair) has been compared with simulated LST (Purple line).

[Figure]

Figure AC4 Comparison of 8-daily temperature between GLDAS and CMA observations averaged for the URS from 2001 to 2018.

7. The title of 4.2.1 was too vague.

**Reply**: We have revised this title in the revised manuscript.

Line 379: "4.2.1 Evaluation of projected precipitation and temperature"

8. Temperature and precipitation were directly from SSP126 and SSP585 dataset. Should they appear before the simulated discharge in section 4.2?

**Reply**: We do this because we have to evaluate the performance of the corrected GCM forcing data in the USR basin before the projection of future climate change.

9. How about the significance level of the trend of each analyzed variable?

**Reply**: The trend significance level of each analyzed variable is determined by Sen's slope (Sen, 1968) and the non-parametric Mann-Kendall (MK) test (Mann, 1945; Kendall, 1975) at a 5% significance level. The "*" in figures of each analyzed variable represents the 95% confidence level and denotes that the trend is statistically significant. We have supplemented the legend for Figs.10, 11 and 15.

10. What's your novelty compared to others' studies?

**Reply**: As can be seen from the Table AC1, the main novelty of this study can be summarized below:

1) For the model simplification and the available scarce observations, most of the studies have utilized existing hydrological models linked with a simple temperature-index model (or day-degree model) for the simulation of snow and glacier melting in the USR basin. However, the hydrological model used in this study not only considers snow and glacier modules based on energy balance, but also coupling the frozen soil parameterization schemes based on energy balance (Shrestha et al., 2010; Wang et al., 2010, 2016, 2017).

2) Previous studies used the CMIP5 dataset, and this study uses the new CMIP6 dataset (ISIMIP3b) that defined by optimal combinations of SSPs and RCPs.

3) Previous studies did not consider the internal snow change processes in the USR basin, which may lead to a partial understanding of the snow-hydrology processes. However, we predicted the change of snowfall, snow cover, SWE, total snowmelt, snowmelt runoff under different SSP scenarios.

Table AC1 Comparison of the major studies in snow hydrological simulations at the USR basin

| Authors | Study periods | Precipitation data source | Future forcing data | Model | Energy-balance snow and glacier module? | Frozen module? | Internal snow processes |
|---------|---------------|---------------------------|---------------------|-------|------------------------------------------|----------------|-------------------------|
| Lutz et al. (2014) | 1998–2007 | APHRODITE | CMIP5 | SPHY | No | No | Snowmelt |
| Su et al. (2016) | 1971–2000; 2011–2040; 2041–2070 | APHRODITE | CMIP5 | VIC-glacier | Yes(snow)/No (glacier) | Yes | Snowmelt |
| Zhao et al. (2019) | 1971–2100 | CMA | CMIP5 | VIC-CAS | Yes(snow)/No (glacier) | Yes | Snowmelt |
| Khanal et al. (2021) | 1979–2100 | ERA5 | CMIP6 | SPHY | No | No | snowmelt |
| Kraaijenbrink et al. (2021) | 1979–2100 | ERA5 | CMIP6 | temperature index (TI) melt model | No | No | SWE, Snowmelt |
| Yang et al. (2021) | 1980–2018 | ERA5, MSWEP | — | GBHM | No | No | Snowmelt |
| Yang et al. (2001) | 1979–2019 | CMA | — | WEP-C | Yes | Yes | Snowmelt |
| This study | 1995-2100 | ERA5 | CMIP6 | WEB-DHM | Yes | Yes | Snowfall, snow cover, SWE, |

| | | | | | | | snowmelt, snow runoff |
|---|---|---|---|---|---|---|---|

11. Some paragraphs were too long. Some errors, such as "would be ere more" in Line 500. Maybe it's better to ask a native English speaker to polish the English before acceptation.

**Reply**: We have done this in the revised manuscript.

**Reference**

1. Ebert, E. E., Janowiak, J. E., and Kidd, C.: Comparison of Near-Real-Time Precipitation Estimates from Satellite Observations and Numerical Models, Bull. Am. Meteorol. Soc., 88, 47-64. https://journals.ametsoc.org/view/journals/bams/88/1/bams-88-1-47.xml, 2007.

2. Kendall, M. G.: Rank Correlation Methods (4th ed.), Charles Griffin, London, 1975.

3. Mao, R. J., Wang, L., Zhou, J., Liu, X. P., Qi, J., and Zhong, X. Y.: Evaluation of Various Precipitation Products Using Ground-Based Discharge Observation at the Nujiang River Basin, China. Water, 11, 2308, https://doi.org/10.3390/w11112308, 2019.

4. Mann, H. B.: Nonparametric tests against trend, Econometrica, 13, 245–259, https://doi.org/10.2307/1907187, 1945.

5. Qi, J., Wang, L., Zhou, J., Song, L., Li, X. P., and Zeng, T.: Coupled Snow and Frozen Ground Physics Improves Cold Region Hydrological Simulations: An Evaluation at the upper Yangtze River Basin (Tibetan Plateau). J. Geophys. Res-Atmos., 124, 12985–13004, https://doi.org/10.1029/2019JD031622, 2019.

6. Sen, P. K.: Estimates of the regression coefficient based on Kendall's Tau, J. Am. Stat. Assoc., 63, 1379–1389, https://doi.org/10.1080/01621459.1968.10480934, 1968.

7. Shrestha, M., Wang, L., Koike, T., Xue, Y., and Hirabayashi, Y.: Improving the snow physics of WEB-DHM and its point evaluation at the SnowMIP sites. Hydrol. Earth Syst. Sci., 14, 2577–2594, https://doi.org/10.5194/hess-14-2577-2010, 2010.

8. Tian, Y., Peters-Lidard, C. D., Eylander, J. B., Joyce, R. J., Huffman, G. J., Adler, R. F., Hsu, K., Turk, F. J., Garcia, M., and Zeng, J.: Component analysis of errors in satellite-based precipitation estimates, J. Geophys. Res., 114, D24101, https://doi.org/10.1029/2009JD011949, 2009.

9. Wang, L., Sun, L., Shrestha, M., Li, X., Liu, W., Zhou, J., Yang, K., Lu, H., and Chen, D.: Improving snow process modeling with satellite-based estimation of near-surface-air-temperature lapse rate, J. Geophys. Res. Atmos., 121, 12005–12030, doi:10.1002/2016JD025506, 2016.

10. Wang, L., Koike, T., Yang, K., Yeh, J. F.: Assessment of a distributed biosphere hydrological model against streamflow and MODIS land surface temperature in the upper Tone River Basin, J. Hydrol., 377: 21–34, https://doi.org/10.1016/j.jhydrol.2009.08.005, 2009b.

11. Wang, L., Koike, T., Yang, K., Jin, R., and Li, H.: Frozen soil parameterization in a distributed biosphere hydrological model, Hydrol. Earth Syst. Sci., 14, 557–571, https://doi.org/10.5194/hess-14-557-2010, 2010.

12. Wang, L., Zhou, J., Qi, J., Sun, L., Yang, K., Tian, L., Lin, Y., Liu, W., Shrestha, M., Xue, Y., Koike, T., Ma, Y., Li, X., Chen, Y., Chen, D. Piao, S., and Lu, H.: Development of a land surface model with coupled snow and frozen soil physics, Water Resour. Res., 53, 5085–5103,

https://doi.org/10.1002/2017WR020451, 2017.

13. Yang, F., Lu, H., Yang, K., Huang, G., Wang, W., Lu, P., Tian, F. and Huang, Y.: Hydrological characteristics and changes in the Nu-Salween River basin revealed with model-based reconstructed data, J. Mt. Sci. 18, 2982–3002. https://doi.org/10.1007/s11629-021-6727-1, 2021.

14. Zhou, J., L. Wang, Y. Zhang, Y. Guo,X. Li, and W. Liu: Exploring thewater storage changes in the largestlake (Selin Co) over the Tibetan Plateauduring 2003–2012 from a basin-wide hydrological modeling, Water Resour.Res., 51, 8060–8086, doi:10.1002/2014WR015846, 2015.

15. Zhong X, Wang L, Zhou J, Li, X. and Wang, Y.: Precipitation Dominates Long-Term Water Storage Changes in Nam Co Lake (Tibetan Plateau) Accompanied by Intensified Cryosphere Melts Revealed by a Basin-Wide Hydrological Modelling, Remote Sens-Base, 12, 1926, https://doi.org/10.3390/rs12121926, 2020.

---

## Author Comment (AC3)

**Reply to Community Comments**

1. How is the snowfall determined? Is the determination of snowfall for future scenarios guaranteed to be accurate?

Reply:

The WEB-DHM-sf model uses a single-temperature threshold method, in which the precipitation below the threshold is considered as snowfall (Shrestha et al., 2014). This method is widely used in hydrologic and land surface models due to its easy access to input data, such as VIC (Liang et al., 1996), SWAT (Arnold et al., 1998) and Mike SHE (Rsfsgaard et al., 1992). Meanwhile, Ding et al. (2014) (about precipitation type identification) illustrated that the single-temperature threshold method has a great applicability in areas where the relative humidity is lower than 78%, and furthermore has a very good performance in high-altitude areas, such as the Tibetan Plateau (TP). Therefore, the accuracy of air temperature and precipitation is a prerequisite for determination of snowfall.

We believe that the reliability of the projected snowfall can be improved by using reliable forcing data (precipitation and temperature), thereby reducing the uncertainty of projected results as much as possible. We need to evaluate and correct the input GCM forcing data to ensure its reliability in the historical period. On this basis, we further ensure the consistency of its future trend. Here, we used 4 bias-corrected ISIMIP 3b GCM (GFDL-ESM4, IPSL-CM6A-LR, MPI-ESM1-2-HR, MRI-ESM2-0) datasets, which have a consistent experimental protocol (historical, SSP126, and SSP585) and atmospheric climate variables (spatial and temporal resolutions of 1 day and 0.5°). To maintain the relative and absolute trends of these variables during the historical and future periods at the basin scale, we further corrected these variables on the monthly scale using the delta method. We then validate the performance of the projected forcing datasets in history periods by comparing the simulated discharges to observations (Fig. 7). With the above efforts, we reduce the uncertainty of the forcing datasets and make the trends of the future projections reasonable.

2. Have snowfall and SWE been verified? They are different from SCA.

Reply:

We didn't perform validations on snowfall and SWE due to a lack of *in-situ* observations at the USR basin. However, the spatial distribution of snow cover area and the snowmelt runoff at the basin outlet implicitly represent the integrated amount of snowfall. Satellite-derived global snow cover area (SCA) by MODIS, known as a reliable snow index for representing large-scale snow variability, is the most effective validation data in hydrologic modelling to quantify the spatial distribution of snow in poorly gauged mountainous river basin (Wang et al., 2009; Shretha et al., 2010, 2014; Zhou et al., 2021). Furthermore, the present snow module considers the attenuation of shortwave radiation penetrated in a three-layer snow pack and the snow-covered surface albedo scheme, and the enthalpy ($H$) is used as a prognostic variable instead of snow temperature in the energy balance equation, which includes the internal energy of liquid water or ice as well as the energy of the phase change. It is assumed that liquid water at its melting point has zero enthalpy so that the phase change processes can be dealt with easily. Therefore, the WEB-DHM-sf model can provide realistic simulation of complex snow physics and have the ability to integrate the measurable physical quantities (Shrestha et al., 2010, 2014; Wang et al., 2017).

3.  What is the calculation of the contribution of snow to the runoff?

Reply:

The WEB-DHM-sf model adopts a three-layer energy balance snow scheme of the Simplified Simple Biosphere 3 model (SSiB3, Xue et al., 2003) and the prognostic albedo scheme of the Biosphere Atmosphere Transfer Scheme (Yang et al., 1997). Within a given subbasin, a number of flow intervals are specified to present time lag and accumulating processes in the river network according to the distance to the outlet of the subbasin, each flow interval includes several model grids. For each model grid with one combination of land use type and soil type, the SSIB3 is used to calculate snow processes, of which the snowmelt can be calculated by the energy balance equations. The amount of snowmelt *Mg/Ms* (m w.e.) of each model grid produced over a period of time can then be expressed as,

$$M_{g/s} = \frac{Q_m}{\rho_w \times h_v} \tag{1}$$

Where $\rho_w$ is the density of liquid water, and the *hv* is the latent heat of fusion; Q_m is

the energy consumed by melt of snow. The energy balance for a ground snowpack or canopy snowpack can be written as:

Ground $\qquad Q_{Mg} = R_n + H + \lambda E + G_{pr} + G_g - \xi$ $\qquad$ (2)

Canopy $\qquad Q_{Mc} = R_n + H + \lambda E + G_{pr} - \xi$ $\qquad$ (3)

Where the subscripts 'g' and 'c' refers to the ground and canopy respectively, Rn (W m$^{-2}$) = Rn$_{sw}$+Rn$_{lw}$ , net radiation which is the sum of net shortwave ($Rn_{sw}$) and longwave radiation ($Rn_{lw}$), H (Wm$^{-2}$) the sensible heat flux exchanged between snow and atmosphere, $\lambda E$ (Wm$^{-2}$) the latent heat flux exchanged between snow and atmosphere, $G_{pr}$ (Wm-2) the sensible heat flux supplied by rainfall, $G_g$ (Wm$^{-2}$) the conductive heat flux exchanged between snow and soil, $\xi$ (J m$^{-2}$) is internal energy of the snowpack, and $Q_{mg}$/$Q_{mc}$ is the energy consumed by melt of snow. Fluxes towards the surface are considered positive and vice versa. These energy interactions are followed by the change in internal energy or the cold content of the snowpack which results change in its temperature and its phase. Melt occurs when the snowpack reaches 0°C and $Q_{mg}$/$Q_{mc}$ is positive. More detailed theories and formulas can be found in Shrestha et al. (2010, 2014) and Wang et al. (2017).

Here, we assume that the longest time of snowmelt flow routing in the basin is less than one month, that is, the snowmelt of all grids reaches the basin outlet within one month. Therefore, the total snowmelt is the accumulation of snowmelt from all grids in the basin (Wang et al., 2009a, b, 2017).

Reference
1. Arnold, J., Srinivasan, R., Muttiah, R., and Williams, J.: Large area hydrologic modeling and assessment part I: Model development. Journal of the American Water Resources Association, 34: 73–89, https://doi.org/10.1111/j.1752-1688.1998.tb05961.x, 1998.
2. Ding, B., Yang, K., Qin, J., Wang, L., Chen, Y., and He, X.: The dependence of precipitation types on surface elevation and meteorological conditions and its parameterization, J. Hydrol., 513: 154–163, https://doi.org/10.1016/j.jhydrol.2014.03.038, 2014.
3. Refsgaard, J., Seth, S., Bathurst, J., Erlich, M., Storm, B., Jørgensen, G., Chandra, S.: Application of the SHE to catchments in India. Part I: General results. J. Hydrol., 140: 1-23, https://doi.org/10.1016/0022-1694(92)90232-K, 1992.
4. Liang, X., Wood, E., and Lettenmaier, D.: Surface soil moisture parameterization

of the VIC-2L model: Evaluation and modification, Global Planet. Change, 13: 195–206, https://doi.org/10.1016/0921-8181(95)00046-1, 1996.

5. Shrestha, M., Wang, L., Koike, T., Xue, Y., and Hirabayashi, Y.: Improving the snow physics of WEB-DHM and its point evaluation at the SnowMIP sites. Hydrol. Earth Syst. Sci., 14, 2577–2594, https://doi.org/10.5194/hess-14-2577-2010, 2010.

6. Shrestha, M., Wang, L., Koike, T., Tsutsui, H., Xue, Y., and Hirabayashi, Y.: Correcting basin-scale snowfall in a mountainous basin using a distributed snowmelt model and remote-sensing data, Hydrol. Earth Syst. Sci., 18, 747–761, https://doi.org/10.5194/hess-18-747-2014, 2014.

7. Wang, L., Koike, T., Yang, K., Jackson, T. J., Bindlish, R., and Yang, D. Development of a distributed biosphere hydrological model and its evaluation with the Southern Great Plains Experiments (SGP97 and SGP99), J. Geophys. Res., 114, D08107, https://doi.org/10.1029/2008JD010800, 2009a.

8. Wang, L., Koike, T., Yang, K., Yeh, J. F.: Assessment of a distributed biosphere hydrological model against streamflow and MODIS land surface temperature in the upper Tone River Basin, J. Hydrol., 377: 21–34, https://doi.org/10.1016/j.jhydrol.2009.08.005, 2009b.

9. Wang, L., Zhou, J., Qi, J., Sun, L., Yang, K., Tian, L., Lin, Y., Liu, W., Shrestha, M., Xue, Y., Koike, T., Ma, Y., Li, X., Chen, Y., Chen, D. Piao, S., and Lu, H.: Development of a land surface model with coupled snow and frozen soil physics, Water Resour. Res., 53, 5085–5103, https://doi.org/10.1002/2017WR020451, 2017.

10. Xue, Y., Sun, S., Kahan, D. S., and Jiao, Y.: Impact of parameterizations in snow physics and interface processes on the simulation of snow cover and runoff at several cold region sites. J. GEOPHYS. RES., 108, 8859. https://doi.org/10.1029/2002JD003174, 2003.

11. Yang, Z. L., Dichinsion, R. E., Robock, A., and Vinnikov, K. Y.: Validation of the snow submodel of the biosphere-atmosphere transfer scheme with Russian snow cover and meteorological observational data. Journal of Climate, 10(2), 353–373. https://doi.org/10.1175/1520-0442(1997)010<0353:VOTSSO>2.0.CO;2, 1997.

12. Zhou, J., Wang, L., Zhong, X., Yao, T., Qi, J., Wang, Y., and Xue, Y.: Quantifying the major drivers for the expanding lakes in the interior Tibetan Plateau, Sci. Bull., 67, 474–478, https://doi.org/10.1016/j.scib.2021.11.010, 2021.